# Autotaxin–lysophosphatidic acid–LPA$_3$ signaling at the embryo-epithelial boundary controls decidualization pathways

Shizu Aikawa[1,†], Kuniyuki Kano[1,2,†], Asuka Inoue[1,3], Jiao Wang[1], Daisuke Saigusa[2,4], Takeshi Nagamatsu[5], Yasushi Hirota[5], Tomoyuki Fujii[5], Soken Tsuchiya[6], Yoshitaka Taketomi[7,8], Yukihiko Sugimoto[2,6], Makoto Murakami[2,7,8], Makoto Arita[9,10], Makoto Kurano[2,11], Hitoshi Ikeda[2,11], Yutaka Yatomi[2,11], Jerold Chun[12] & Junken Aoki[1,2,*]

## Abstract

During pregnancy, up-regulation of heparin-binding (HB-) EGF and cyclooxygenase-2 (COX-2) in the uterine epithelium contributes to decidualization, a series of uterine morphological changes required for placental formation and fetal development. Here, we report a key role for the lipid mediator lysophosphatidic acid (LPA) in decidualization, acting through its G-protein-coupled receptor LPA$_3$ in the uterine epithelium. Knockout of *Lpar3* or inhibition of the LPA-producing enzyme autotaxin (ATX) in pregnant mice leads to HB-EGF and COX-2 down-regulation near embryos and attenuates decidual reactions. Conversely, selective pharmacological activation of LPA$_3$ induces decidualization via up-regulation of HB-EGF and COX-2. ATX and its substrate lysophosphatidylcholine can be detected in the uterine epithelium and in pre-implantation-stage embryos, respectively. Our results indicate that ATX–LPA–LPA$_3$ signaling at the embryo-epithelial boundary induces decidualization via the canonical HB-EGF and COX-2 pathways.

**Keywords** autotaxin; decidualization; embryo implantation; LPA$_3$; lysophosphatidic acid

**Subject Categories** Development & Differentiation; Signal Transduction

The EMBO Journal (2017) 36: 2146–2160

## Introduction

Infertility is a global problem experienced by about 10–15% of couples during their reproductive years. In addition, in spite of recent advances in the artificial reproductive technology (ART), the success rate of pregnancy through ART is still low (~30%; Lim & Wang, 2010; Ramathal *et al*, 2010; Cha *et al*, 2012). These pregnancy failures are believed to be mainly due to defects in early pregnancy events including implantation and decidualization.

Decidualization is a series of uterine morphological changes during early pregnancy which is essential for the following placental formation and fetal development (Lim & Wang, 2010; Ramathal *et al*, 2010; Cha *et al*, 2012). In mice, decidualization occurs only in the vicinity of the embryos. In humans, this process occurs cyclically, whether or not an embryo is present (pre-decidualization), although it is reinforced by embryo implantation (Cha *et al*, 2012). In the decidual process, the uterine epithelium breaks down, and extensive proliferation and angiogenesis occur in the subepithelial stroma (decidual reactions). The signaling required for proper decidualization is generated through two types of cell–cell interactions, i.e., embryo-epithelial and epithelial–stromal interactions. Maternal factors such as heparin-binding epidermal growth factor (HB-EGF) and cyclooxygenase-2 (COX-2) are induced in the epithelium surrounding the embryos (embryo-epithelial interaction) and then act on the stroma to induce the expression of bone morphogenic protein 2 (Bmp2) and wingless-related MMTV integration site 4

---

1 Graduate School of Pharmaceutical Sciences, Tohoku University, Sendai, Miyagi, Japan
2 Japan Agency for Medical Research and Development, Core Research for Evolutional Science and Technology (AMED-CREST), Chiyoda-ku, Tokyo, Japan
3 Japan Science and Technology Agency, Precursory Research for Embryonic Science and Technology (PRESTO), Kawaguchi, Saitama, Japan
4 Department of Integrative Genomics, Tohoku Medical Megabank, Tohoku University, Sendai, Miyagi, Japan
5 Department of Obstetrics and Gynecology, Faculty of Medicine, The University of Tokyo, Bunkyo-ku, Tokyo, Japan
6 Department of Pharmaceutical Biochemistry, Graduate School of Pharmaceutical Sciences, Kumamoto University, Kumamoto, Japan
7 Tokyo Metropolitan Institute of Medical Science, Setagaya-ku, Tokyo, Japan
8 Center for Disease Biology and Integrative Medicine, Graduate School of Medicine, The University of Tokyo, Bunkyo-ku, Tokyo, Japan
9 RIKEN, Center for Integrative Medical Sciences, Yokohama, Kanagawa, Japan
10 Graduate School of Pharmaceutical Sciences, Keio University, Minato-ku, Tokyo, Japan
11 Department of Clinical Laboratory, The University of Tokyo Hospital, Bunkyo-ku, Tokyo, Japan
12 Sanford Burnham Prebys Medical Discovery Institute, La Jolla, CA, USA
  *Corresponding author. Tel: +81 22 795 6860; Fax: +81 22 795 6859; E-mail: jaoki@m.tohoku.ac.jp
  †These authors contributed equally to this work

(Wnt4) in stroma (epithelial–stromal interaction; Lim *et al*, 1997; Paria *et al*, 2001; Song *et al*, 2002; Wang *et al*, 2004; Xie *et al*, 2007; Large *et al*, 2014). Mice in which these maternal factors (HB-EGF, COX-2, Bmp2, and Wnt4) are knocked out showed obvious defects in decidual reactions (Lim *et al*, 1997; Wang *et al*, 2004; Lee *et al*, 2007; Xie *et al*, 2007; Franco *et al*, 2011; Li *et al*, 2013; Large *et al*, 2014). Based on these observations, it has been proposed that some factor(s) are present in the embryo-epithelial boundary and induce the expression of HB-EGF and COX-2 in the epithelium, which in turn facilitate the following decidual reactions through up-regulation of Bmp2 and Wnt4 (Paria *et al*, 2001; Cha *et al*, 2012).

Lysophosphatidic acid (LPA) has various roles as a lipid mediator through G-protein-coupled receptors (GPCR). So far, six LPA-specific GPCRs named LPA₁-LPA₆ have been identified (Aikawa *et al*, 2015; Sheng *et al*, 2015). LPA is mainly synthesized from lysophospholipids such as lysophosphatidylcholine (LPC) by a secretory enzyme autotaxin (ATX; Umezu-Goto *et al*, 2002). Studies of knock-out (KO) mice and human genetic diseases of these LPA-related genes have shown that LPA has various pathophysiological roles including angiogenesis (Yukiura *et al*, 2011), hair follicle formation (Inoue *et al*, 2011; Hayashi *et al*, 2015), bone development (Nishioka *et al*, 2016), and neural development (Yung *et al*, 2015). We previously showed that LPA₃, which is highly expressed in the uterine epithelium during the peri-implantation period (Ye *et al*, 2005), has a critical role in the early pregnancy. *Lpar3* KO mice show many reproductive defects, including significantly reduced COX-2 (a key enzyme for synthesis of prostaglandins), delayed implantation, aberrant embryo spacing, defects in placental formation and fetal development, and reduced litter size (Ye *et al*, 2005; Hama *et al*, 2007). However, the defects in *Lpar3* KO were only partially recovered by administration of prostaglandins (Ye *et al*, 2005), suggesting that other unidentified factors should operate downstream of LPA₃. In addition, it is almost unclear what kind of cellular events are affected in *Lpar3* KO uteri.

In this study, to gain insights into the signaling and cellular events downstream of LPA₃, we administered a potent agonist for LPA₃ into the mouse uterine cavity during the peri-implantation period. Unexpectedly, mere activation of the epithelial LPA₃ by the agonist induced prominent endometrial morphological changes, which were associated with up-regulation of the above-mentioned decidual factors (HB-EGF, COX-2, Bmp2, and Wnt4). Furthermore, we obtained evidences that endogenously LPA₃ signaling was evoked by ATX, an LPA-producing enzyme. These results lead us to propose a novel mechanism for decidualization elicited by embryos; that is, the ATX–LPA₃ axis in the embryo-epithelial boundary regulates decidualization by inducing maternal factors such as HB-EGF and COX-2.

## Results

### An LPA₃ agonist, T13, induces decidualization

To clarify the molecular mechanisms and cellular events induced downstream of LPA₃ signaling, we injected T13, a potent LPA₃ agonist (EC₅₀ ~0.2 nM; Fig EV1A–C; Tamaruya *et al*, 2004; Kano *et al*, 2008; Hama & Aoki, 2010), into the uterine cavities of pseudopregnant mice at 3.5 days post-coitus (dpc). Interestingly, T13 induced dramatic uterine hypertrophy throughout the uterine horns at 5.5 dpc (Figs 1A and B, and 2A). The T13-induced hypertrophy was completely absent in the uteri of *Lpar3* KO uteri mice (Figs 1 and 2A), indicating T13 evokes uterine hypertrophy through the activation of LPA₃. T13 induced several cellular changes, which resembled the changes that occur during decidual reactions in normal pregnancy. At 4.5 dpc, stromal proliferation as judged by bromodeoxyuridine (BrdU) labeling was evident in the stromal cells surrounding the embryo (primary decidual zone; PDZ; Fig 2B, upper row). At 5.5 dpc, the proliferative area expanded outside the

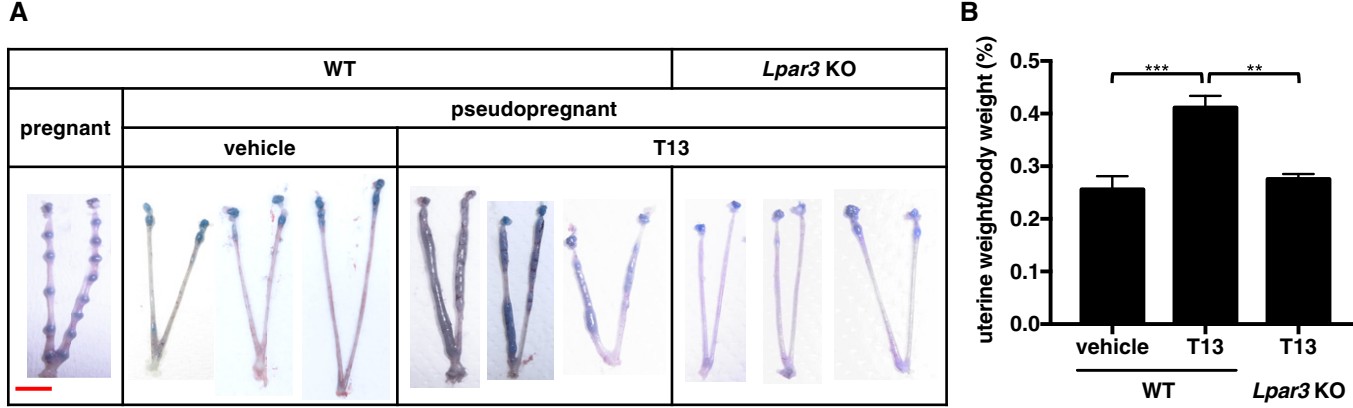

**Figure 1.   Intrauterine injection of a potent LPA₃ agonist, T13, causes uterine hypertrophy.**

A, B   Representative photographs of pregnant or T13-injected pseudopregnant uteri on 5.5 dpc (A) and the average mass of uteri (B, *n* = 5 for WT with vehicle and *Lpar3* KO with T13, *n* = 7 for WT with T13). At 3.5 dpc, vehicle or an LPA₃ agonist (T13) was injected into pseudopregnant uteri and mice were dissected at 5.5 dpc. In the normal pregnant uterus, only implantation sites were enlarged because of decidualization. On the other hand, pseudopregnant uteri showed no enlargement since they had no blastocysts in the luminal cavity. The injection of T13 into pseudopregnant uteri caused dramatic hypertrophy throughout the uterine horns in an LPA₃-dependent manner. Each image in (A) is a representative from at least three independent experiments. Scale bar = 1 cm. Data are means + SEM. **$P < 0.01$, ***$P < 0.001$ by ANOVA.

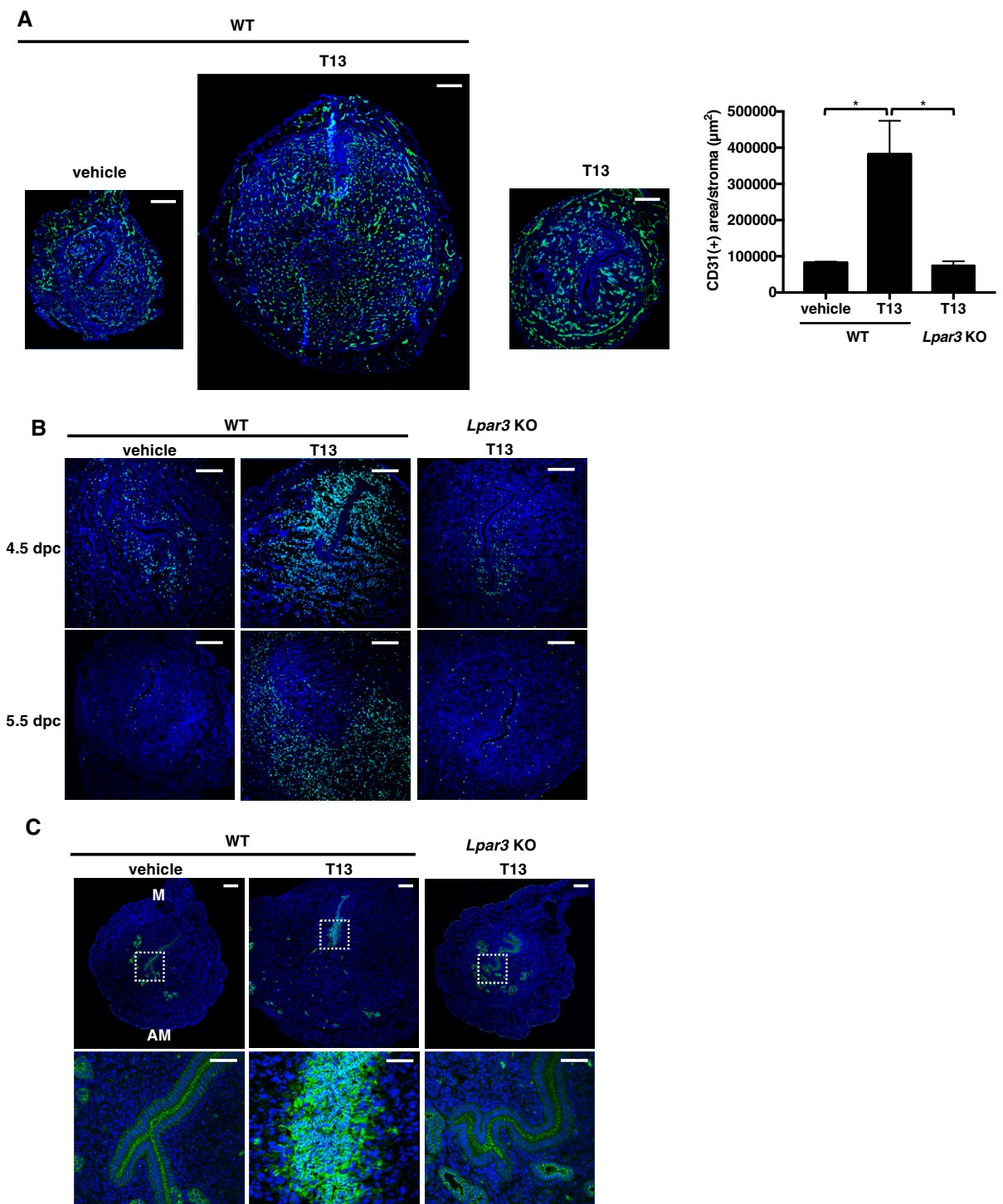

**Figure 2.  Activation of LPA₃ evokes decidual reactions.**

A–C  Immunostaining of CD31 (angiogenesis, A), BrdU (cell proliferation, B), and E-cadherin (LE-breakdown, C) in T13-treated pseudopregnant uteri showed that T13 induced prominent decidual reactions in an LPA₃-dependent manner. (A) In T13-injected uteri on 5.5 dpc, fine vascular formation was observed in the AM pole. The CD31 (+) area was calculated by ImageJ (right, $n = 3$ for WT with vehicle and *Lpar3* KO with T13, $n = 4$ for WT with T13). Data are means + SEM. *$P < 0.01$ by ANOVA. (B) T13-treated mice were injected with BrdU on the morning of 4.5 or 5.5 dpc. After 2-h chasing, uteri were dissected. Cell proliferation occurred in the PDZ on 4.5 dpc and then SDZ on 5.5 dpc. Higher magnification images of boxed regions are shown in the lower panels. (C) T13 caused LE-breakdown in the AM pole on 5.5 dpc. In each image, nuclei were counterstained with DAPI (blue). M, mesometrial pole; AM, anti-mesometrial pole. Scale bar: 200 μm (A, B and upper row in C) and 50 μm (lower low in C). Each image is a representative from at least three independent experiments.

Source data are available online for this figure.

    

**Table 1.  Numbers of genes up- and down-regulated by at least a factor of two in uteri treated with T13.**

| Type of regulation | No. of genes regulated at time after T13 injection (h) | | |
|---|---|---|---|
| | **0.5** | **1** | **2** |
| Up-regulated[a] | 10 | 10 | 29 |
| Down-regulated[b] | 7 | 8 | 12 |

The data have been deposited in the NCBI GEO database (accession no. GSE87116). The following genes are up- or down-regulated in 2 h after the T13 injection. *Hbegf* and *Ptgs2* (in bold), responsible genes for uterine functions, were up-regulated.

[a]*Mup1, Cftr, Vmn1r12/Vmn1r14, Gjb2, A2m,* **Ptgs2**, *Vmn2r88, P2yr14, Edn1, Vnn1, Ly6a, Cyp4a11, Gpcr5a, Hsd3b1, Csn2, Ifi16, Cyp3a25, Rcan1, Ramp3, Lcn2, Crtac1, mir-15,* **Hbegf**, *Scl40s1, Ighm, Gadd45 g, Sirpb1, Dip2, Egr3.*
[b]*Lpin2, Tp53i11, Rnasel, Mcpt4, Ccl5, Sult5a1, Tac3, Ceacam1, Cxcl10, Cxcl9, Igk-v28, H2-t22.*

PDZ (Fig 2B, lower row). In addition, angiogenesis as judged by anti-CD31 staining was prominent in the stromal layer (Fig 2A). At this time, the luminal epithelium collapsed (LE-breakdown) at the antimesometrial (AM) pole, as shown by E-cadherin staining in T13-treated uteri (Fig 2C). We also confirmed that T13-injected uteri showed high alkaline phosphatase activity which is an indicator of decidualized stromal cells (Appendix Fig S1). LPA₃ activation seems to induce some factor(s) in the epithelial layer, which then evoke the decidual reactions in the stromal layer. It should be noted that oil-induced decidualization was similarly observed both in wild-type and *Lpar3* KO uteri (Fig EV2), confirming that the intrinsic mechanism for decidualization was not affected in *Lpar3* KO uteri. This suggests that LPA does not induce decidualization directly but contributes to the induction of decidualization by up-regulating some decidual factors via LPA₃. Accordingly, we concluded that all

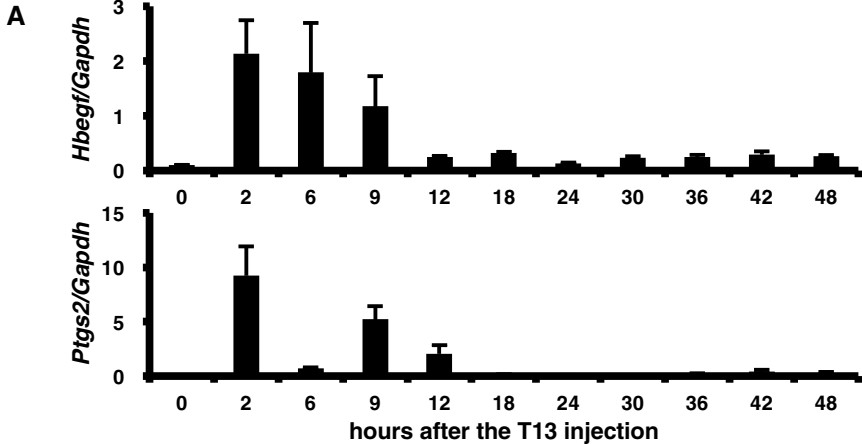

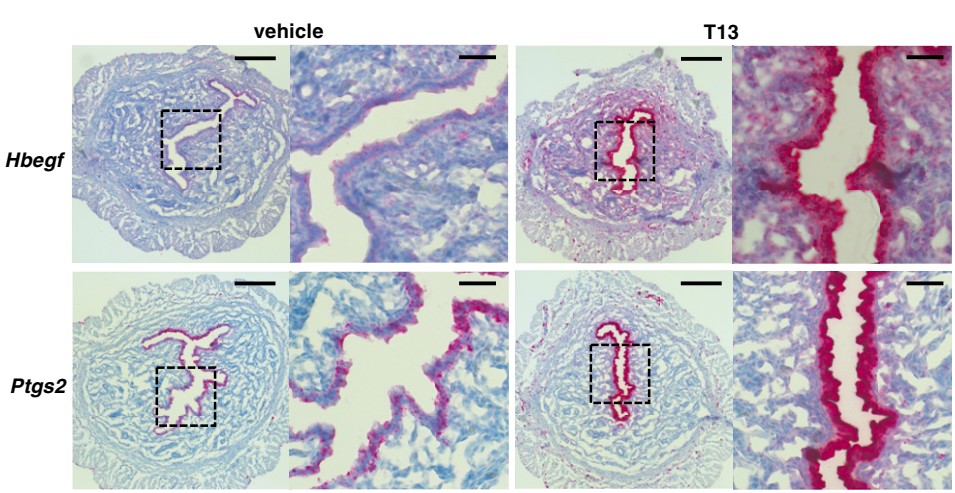

**Figure 3.  HB-EGF and COX-2 were highly induced on the epithelial layer in the T13-injected uteri.**
Temporal and spatial expression of *Hbegf* and *Ptgs2* mRNAs in T13-injected uteri.
A    Time course of qRT–PCR quantification of *Hbegf* and *Ptgs2* mRNAs in T13-injected pseudopregnant uteri (*n* = 8 for 2 h, *n* = 4 for 6 and 9 h, *n* = 5 for 0 and 12–48 h of *Hbegf*; *n* = 12 for 6 h, *n* = 4 for 9 h, *n* = 5 for 0, 2 and 12–48 h of *Ptgs2*). Both *Hbegf* and *Ptgs2* were transiently up-regulated after the treatment. Data are means + SEM.
B    Representative ISH images of *Hbegf* and *Ptgs2* 2 h after the injection. T13 strongly induced both transcripts in the epithelial layer. Higher magnification images of boxed regions are shown in each right panel. Each image is a representative from at least three independent experiments. Scale bar: 200 μm (each left panel) and 50 μm (each right panel).

the decidual reactions (LE-breakdown, stromal proliferation, and angiogenesis) could be induced in the absence of embryos solely by activating LPA$_3$.

## LPA$_3$ evokes decidual reactions through up-regulation of HB-EGF, COX-2, and Bmp2/Wnt4 signalings

To understand the molecular mechanism underlying T13-induced decidualization, we then performed cDNA microarray analyses. Several tens of genes were up-regulated 2 h after the T13 injection (Table 1). Among the genes, we focused on *Hbegf* and *Ptgs2* (encoding COX-2), because they are responsible for decidualization and knockout of these genes interfered with implantation (Lim *et al*, 1997; Song *et al*, 2002; Wang *et al*, 2004; Xie *et al*, 2007; Large *et al*, 2014), as was observed in *Lpar3* KO mice (Ye *et al*, 2005). Both *Hbegf* and *Ptgs2* were transiently induced, peaking at 2–9 h after the T13 injection (Fig 3A). Both genes were predominantly up-regulated in the epithelial layer (Fig 3B). Among the EGF family members, only *Hbegf* was up-regulated by T13 (Fig EV3). In agreement with the up-regulation of *Ptgs2*, a lipidomics analysis confirmed the increase of PGE$_2$ and PGF$_{2\alpha}$ in T13-injected uteri 9 h after the injection. The increase of PGE$_2$ and PGF$_{2\alpha}$ were markedly suppressed by the COX-2 inhibitor (Fig EV4), suggesting that these PGs are involved in T13-induced decidualization.

The microarray analysis also revealed that the expressions of several thousand genes (3,220 genes at 24 h, 3,365 genes at 30 h, and 3,949 genes at 36 h) differed by a factor of at least two between T13-injected uteri and the control (Table 2). Ingenuity pathway analysis (IPA) revealed that genes involved in the cell cycle and DNA replication were highly affected by T13 injection (Table 3), which is in agreement with the observation that T13 induced the uterine hypertrophy. We also found that downstream of LPA$_3$, Bmp2/Wnt4 signaling (Table 2 and Fig 4A) was activated. Bmp2 and Wnt4 are well-known decidual factors acting downstream of HB-EGF signaling (Paria *et al*, 2001; Lee *et al*, 2007; Franco *et al*, 2011; Li *et al*, 2013; Large *et al*, 2014). Interestingly, a significant negative correlation of gene expression profiles was observed between T13-injected uteri and uteri null for either *Egfr* (a target of HB-EGF), *Bmp2,* and *Wnt4* (Large *et al*, 2014; Fig 4B). By upstream analysis using the IPA, we also found a significant positive correlation between genes affected by LPA$_3$ and E$_2$ signaling, the latter of which is important for the endometrial proliferation and the establishment of early pregnancy (Lim & Wang, 2010; Ramathal *et al*, 2010; Cha *et al*, 2012; Pawar *et al*, 2015; Table 4). Conversely, there was a negative correlation between genes affected by LPA$_3$ and an ERα antagonist (fulvestrant; Table 4). These results are consistent with the recent observation that E$_2$ signaling is down-regulated in pregnant *Lpar3* KO uteri (Diao *et al*, 2015).

We next performed *in vivo* experiments to evaluate each signal for the development of T13-induced decidualization. Administration of an EGFR inhibitor (AST1306) reduced the levels of T13-induced *Bmp2* and *Wnt4* (Fig 4B). A COX-2 selective inhibitor (Celecoxib) also down-regulated both *Bmp2* and *Wnt4* (Fig 4B), suggesting that, in addition to HB-EGF signaling, epithelial COX-2 signaling is required for the induction of Bmp2 and Wnt4 in the stromal layer. By contrast, fulvestrant decreased the *Bmp2* level, but not *Wnt4* level (Fig 4B). As expected, neither AST1306 nor Celecoxib decreased the expression of *Hbegf* and *Ptgs2* themselves

**Table 2. Gene expression profiling of uteri treated with T13.**

| Parameter | Time after T13 injection (h) | | |
|---|---|---|---|
| | **24** | **30** | **36** |
| No. genes up-regulated > 2 fold | 776 | 903 | 1,341 |
| No. genes down-regulated > 2 fold | 2,444 | 2,462 | 2,608 |
| *Bmp2* fold increase | 2.6↑ | 6.3↑ | 11.9↑ |
| *Wnt4* fold increase | 1.3↑ | 2.8↑ | 12.2↑ |

The data have been deposited in the NCBI GEO database (accession no. GSE87161).

**Table 3. Top five Molecular and Cellular Functional annotations for differently expressed genes between T13 vs. vehicle-treated uteri 24 h after the injection.**

| Annotation | No. genes |
|---|---|
| Cell cycle | 282 |
| Cellular assembly and organization | 98 |
| DNA replication, recombination, and repair | 215 |
| Molecular transport | 200 |
| Cell-to-cell signaling and interaction | 247 |

Functional annotations were analyzed with Ingenuity Pathway Analysis (IPA) software (Qiagen, CA, USA).

(Appendix Fig S2). Likewise, fulvestrant failed to affect the expression of *Hbegf* and *Ptgs2* (Appendix Fig S2), indicating that LPA$_3$ signaling in the epithelial layer was not affected by E$_2$. The signaling pathways identified above can be inhibited by several factors, including an EGFR inhibitor (AST1306), a COX-2 inhibitor (Celecoxib), an ERα antagonist (fulvestrant), a BMPR inhibitor (LDN193189), or a β-catenin inhibitor (XAV939). Each of these factors strongly suppressed T13-induced decidual events including uterine hypertrophy (Fig 5A), stromal cell proliferation (Fig 5B), angiogenesis (Fig EV5), and LE-breakdown (Fig EV6). These results demonstrate that all these signals were essential for T13-induced decidualization.

## ATX–LPA$_3$ axis endogenously contributes to decidualization

To know the role of endogenous LPA$_3$ signaling in decidualization, we looked into the decidual reactions in *Lpar3* KO uteri. Consistent with the data using the LPA$_3$ agonist, stromal cell proliferation in pregnant *Lpar3* KO uteri was markedly reduced as judged by incorporation of BrdU (Fig 6A and B) and the number of nuclei in the stromal layer (Fig 6C). In addition, *Lpar3* KO uteri showed weakened angiogenesis (Fig 6D and E). LE-breakdown was rarely observed in *Lpar3* uteri (Fig 6F). All these data suggested that endogenous LPA$_3$ signaling is important for the development of decidualization. The expressions of *Hbegf*, *Ptgs2*, *Bmp2,* and *Wnt4* were also concomitantly reduced (Fig 6G and H), which further explains the reduced decidual reactions in *Lpar3* KO uteri.

A remaining question is how LPA is produced in the vicinity of the embryos. It is possible that LPA is derived from embryos and/or uteri. Blastocysts null for either *Enpp2* [encoding ATX] or *Liph* (encoding phosphatidic acid-preferential phospholipase A$_{1\alpha}$ (PA-PLA$_{1\alpha}$,

## A

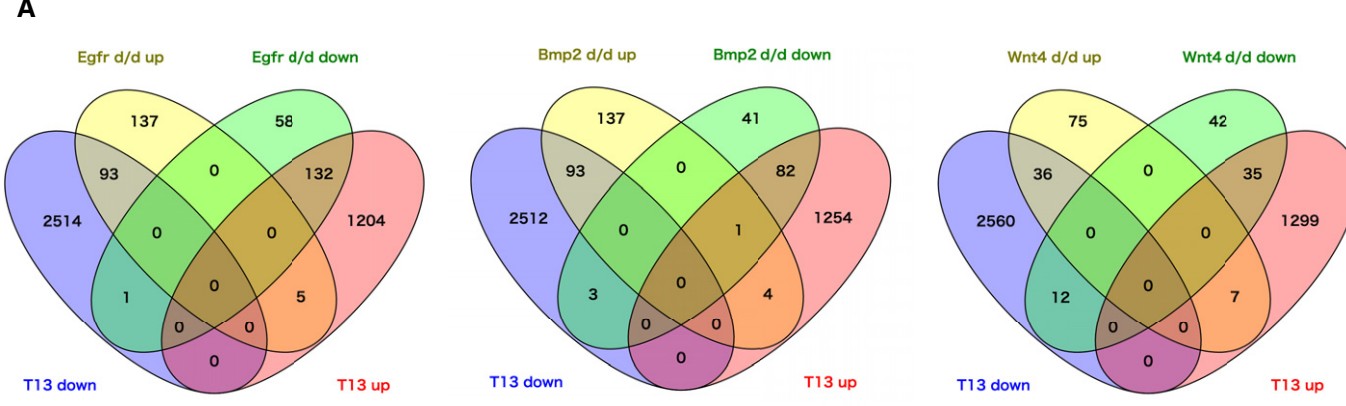

## B

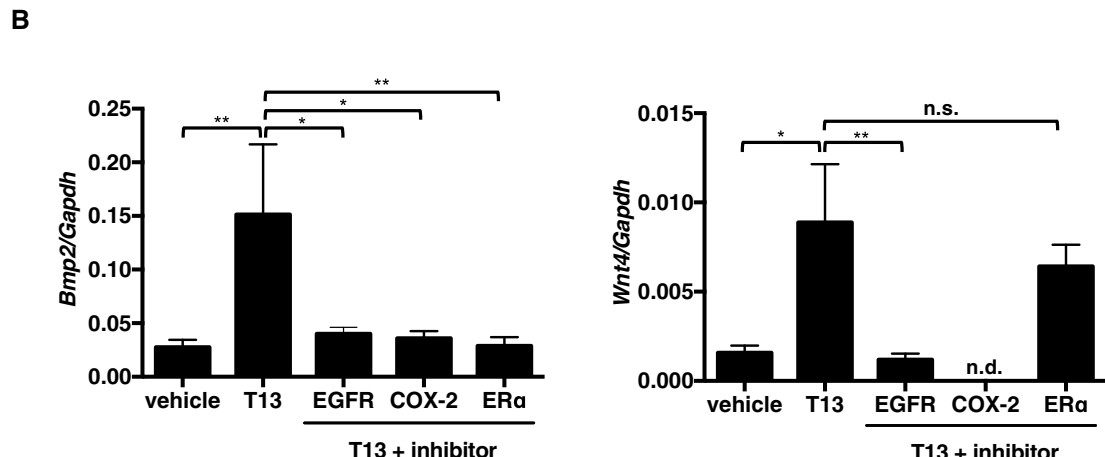

**Figure 4.  Bmp2 and Wnt4 were up-regulated in the T13-injected uteri, downstream of EGFR, COX-2, and ERα signals.**

A  Venn diagrams comparing gene expression signatures between T13-injected uteri (36 h after the injection) and uteri from either KO of *Egfr*, *Bmp2*, or *Wnt4* (24 h after the artificial decidual stimuli). A significant negative correlation in the gene expression patterns was observed between T13-injected and each KO uteri.

B  qRT–PCR quantification of *Bmp2* and *Wnt4* mRNAs in T13-injected uteri 36 h after the treatment ($n = 8$ for vehicle, $n = 7$ for T13, $n = 8$ for T13 + EGFR inhibitor, $n = 10$ for T13 + COX-2 inhibitor, and $n = 9$ for T13 + ERα antagonist). Pharmacological inhibition of EGFR, COX-2 or ERα prominently reduced T13-induced *Bmp2* and *Wnt4* expressions. Data are means + SEM. *$P < 0.05$, **$P < 0.01$, n.s.: not significant by ANOVA. n.d.: not detected. See also Tables 1–4.

another LPA-producing enzyme)] implanted normally (Tanaka *et al*, 2006; Inoue *et al*, 2011), suggesting that embryonic ATX and PA-PLA₁α are not involved in implantation. In addition, *Liph* KO female mice showed normal reproductive activity (Inoue *et al*, 2011). By contrast, the role of maternal ATX remains to be determined, since *Enpp2* null mice were embryonic lethal (Tanaka *et al*, 2006). Anti-ATX staining showed that ATX was predominantly expressed in the uteri and localized throughout the epithelial layer (Fig 7A). Administration of an ATX inhibitor (S15-00826, Fig EV7A–E) into the uterine cavity of pregnant mice resulted in abnormal embryo spacing and implantation failure at 5.5 dpc (Fig 7B–D), as was observed in *Lpar3* KO uteri (Ye *et al*, 2005; Hama *et al*, 2007). The expression levels of *Hbegf*, *Ptgs2*, *Bmp2* and *Wnt4* were also reduced by the ATX inhibitor (Fig 7E and F). These results indicate that epithelially expressed ATX is responsible for the activation of LPA₃ during the peri-implantation period.

## Discussion

Our results indicate that LPA is produced in a maternal ATX-dependent manner and present in the vicinity of the embryo, then activates LPA₃ in the epithelial layer (Fig 8). We tried to detect LPA in the eggs as well as in the luminal fluid using LC-MS/MS. While we could not detect LPA in the eggs, small amount of LPA (0.1–0.2 nM) was found in the uterine flushing fluids from the pregnant mice (Appendix Fig S3). Interestingly, LPA with an unsaturated fatty acid (oleic or linoleic acid), a potent ligand for LPA₃ (Bandoh *et al*, 2000), was detected when the uteri were flushed with the saline containing albumin which is capable of extracting lysophospholipids from outer leaflet of the cells (Okudaira *et al*, 2014; Appendix Fig S3). LPA was hardly recovered in the albumin-free flushing fluids (Appendix Fig S3), indicating clearly that LPA is present in the extracellular milieu. The estimated egg volume is $\sim 6 \times 10^{-14}$ m³

**Table 4. Upstream analysis 24 h after the T13 injection.**

| Upstream regulator | Molecule type | Predicted activation state | *P*-value of overlap |
|---|---|---|---|
| CSF2 | Cytokine | ↑ | 4.43E-21 |
| EP400 | Other | ↑ | 1.95E-14 |
| E2F1 | Transcription regulator | ↑ | 5.49E-13 |
| E2f | Group | ↑ | 4.18E-10 |
| Vegf | Group | ↑ | 2.50E-09 |
| **Estrogen** | **Chemical drug** | ↑ | **4.82E-08** |
| Cephaloridine | Chemical drug | ↑ | 2.29E-07 |
| FOXM1 | Transcription regulator | ↑ | 8.46E-07 |
| CDKN1A | Kinase | ↓ | 1.91E-23 |
| 1-alpha, 25-dihydroxy vitamin D3 | Chemical drug | ↓ | 3.96E-20 |
| Rb | Group | ↓ | 2.07E-13 |
| IRGM | Other | ↓ | 1.17E-12 |
| **Fulvestrant** | **Chemical drug** | ↓ | **2.60E-10** |
| SMARCB1 | Transcription regulator | ↓ | 1.13E-08 |
| PTF1A | Transcription regulator | ↓ | 4.94E-08 |
| BNIP3L | Other | ↓ | 1.08E-07 |

Biological networks and pathways were analyzed with Ingenuity Pathway Analysis (IPA) software (Qiagen, CA, USA). Estrogen signaling was significantly affected by T13 injection (in bold).

(provided that the diameter of the egg is 50 μm), while the volume of uterine cavity is ~$5 \times 10^{-9}$ m³: i.e., the approximate ratio of them = $1:10^5$. Assuming that LPA is produced only in the embryo-epithelial boundary, we can estimate that a high concentration of LPA enough to activate LPA₃ (normally μM order) is present there. Since ATX is present almost exclusively in the epithelial layer (Fig 7A), ATX substrates should be either derived from embryos or produced as a result of embryo-epithelial interactions. Interestingly, lysophosphatidylcholine (LPC) species with an unsaturated fatty acid (oleic or linoleic acid), one of the preferable ATX substrates (Nishimasu *et al*, 2011), were detected in blastocysts on 3.5 dpc by LC-MS/MS (Appendix Fig S4). This is consistent with a report that the luminal epithelium surrounding embryos at 3.5 dpc was rich in unsaturated fatty acid-containing phosphatidylcholine (Burnum *et al*, 2009), the most likely precursor of unsaturated LPC. These data raise the possibility that embryo-derived LPC is converted to LPA by the epithelial ATX, leading to the activation of LPA₃.

The present results also demonstrate a novel mechanism for GPCR-induced cell growth in which LPA₃ is the key player. In our results, activation of LPA₃ in the epithelial layer leads to enhanced production of HB-EGF and COX-2, which then induces stromal cell growth in an EGFR-Bmp2/Wnt4-dependent manner. GPCRs such as β1, AT1, and LPA₆ were previously found to be co-expressed with EGFR in epithelial cells and to activate (actually transactivate) EGFR by stimulating the ectodomain-shedding of membrane bound EGF precursors (Zhai *et al*, 2006; Noma *et al*, 2007; Inoue *et al*, 2011). Interestingly, activation of LPA₃ also induced ectodomain-shedding of HB-EGF *in vitro* (Appendix Fig S5), which indicates that the LPA₃ signaling leads to the activation of EGF signaling not only by up-regulating the expression of HB-EGF but also by ectodomain-shedding of HB-EGF and the following transactivation of EGFR.

The expression of LPA₃ in female reproductive tissues is conserved in mammals. Indeed, in mouse, sheep, pig and human, LPA₃ was expressed in the uterine epithelial layer in a female sex hormone-dependent manner (Hama *et al*, 2006; Kamińska *et al*, 2008; Liszewska *et al*, 2012; Guo *et al*, 2013). In addition, ATX and LPA were detected in the reproductive biological fluids such as follicular fluids and uterine luminal fluids (Liszewska *et al*, 2009; Seo *et al*, 2012; Yamamoto *et al*, 2016). Thus, LPA₃ appears to regulate the female reproductive systems in wide range of mammalian species including human, although there are some slight differences in the process of decidualization between species (Cha *et al*, 2012).

---

**Figure 5.  LPA₃ facilitates decidualization through HB-EGF, COX-2, Bmp2/Wnt4, and ERα signaling pathways.**

A Representative photographs of pseudopregnant uteri treated with T13 and either one of inhibitors (left) and the average uterine weight (right) on 5.5 dpc were shown. Pharmacological inhibition of each signaling molecule suppressed T13-induced hypertrophy (*n* = 7 for vehicle, *n* = 8 for T13, T13 + Wnt/β-catenin inhibitor and T13 + ERα antagonist, *n* = 9 for T13 + EGFR inhibitor and T13 + BMPR inhibitor, *n* = 10 for T13 + COX-2 inhibitor).

B Representative images of BrdU immunostaining in pseudopregnant uteri treated with T13 and either one of inhibitors (2 h BrdU chasing on 5.5 dpc) (left) and the average BrdU (+) counts (right, *n* = 3 for each bar). T13-induced stromal cell proliferation was canceled by treatment of each inhibitor. In each image, nuclei were counterstained with DAPI (blue). The counts of BrdU were calculated using ImageJ.

Data information: Each image in both (A) and (B) is a representative from at least three independent experiments. Scale bar: 1 cm (A) and 200 μm (B). Data are means + SEM, ***P < 0.001, ****P < 0.0001 by ANOVA.

Source data are available online for this figure.

▶

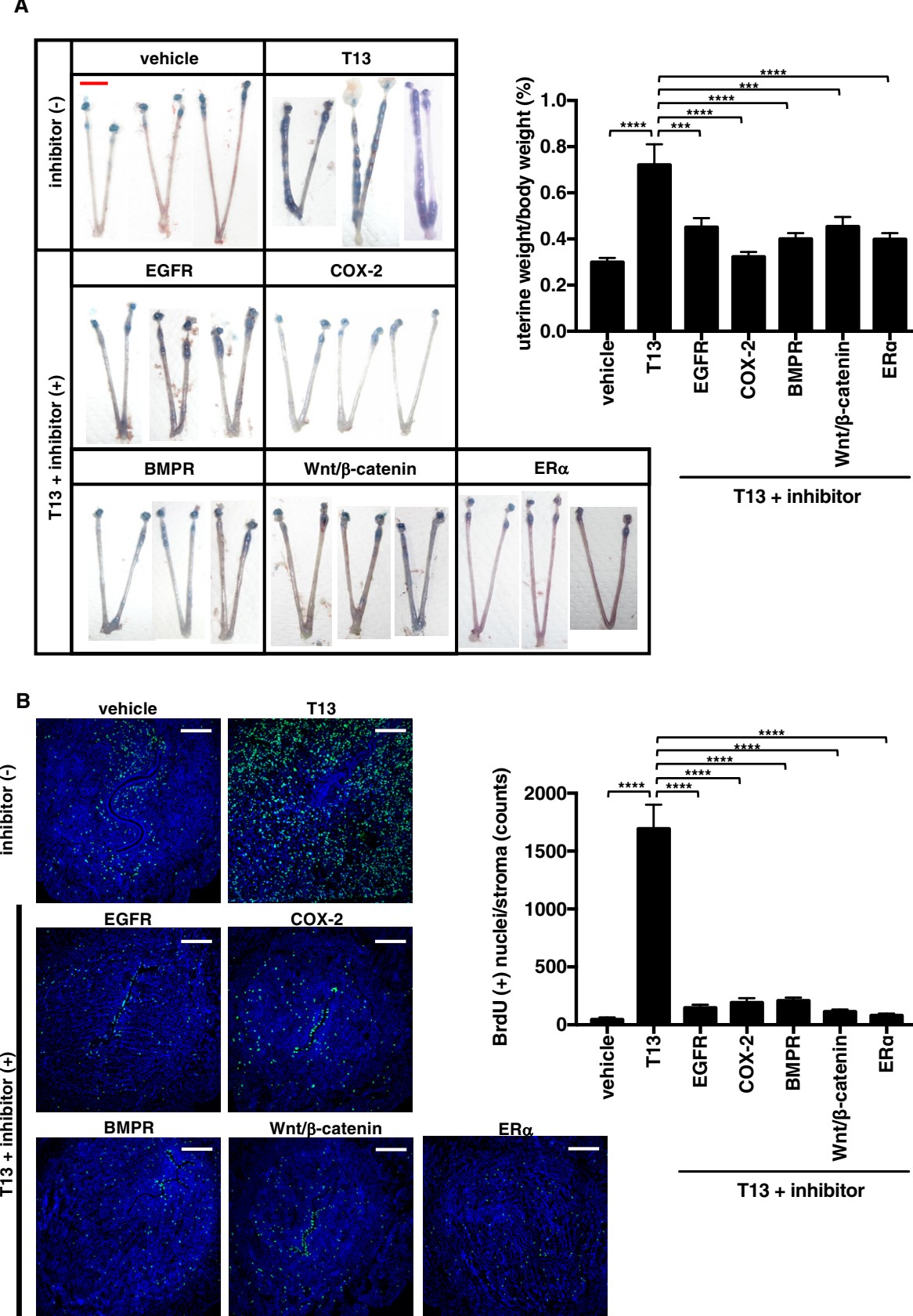

**Figure 5.**

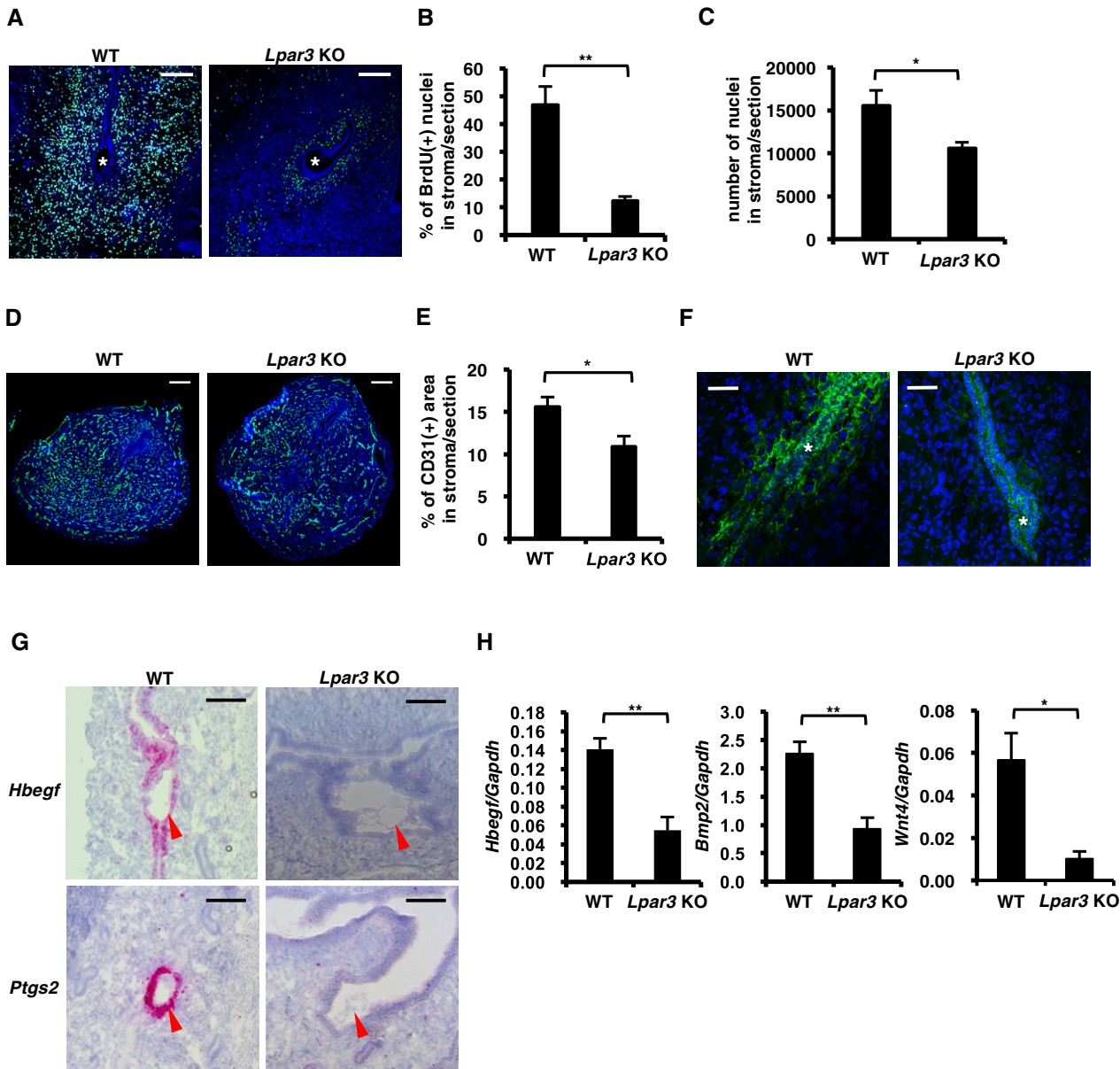

**Figure 6. Endogenous LPA₃ signaling regulates decidual reactions in normal pregnancy.**

A–F  (A, D and F) Immunostaining of BrdU (cell proliferation on 4.5 dpc, A), CD31 (angiogenesis on 5.5 dpc, D), and E-cadherin (LE-breakdown on 5.5 dpc, F) in pregnant uteri showed that decidual reactions were poorly occurred in *Lpar3* KO mice. For BrdU IHC, pregnant mice were treated with BrdU 2 h before the dissection. In each image, nuclei were counterstained with DAPI (blue). Asterisks indicate the position of embryos. (B) Number of BrdU (+) nuclei in the stromal layer per section in (A) was counted by ImageJ software (*n* = 5 for each bar). (C) Number of nuclei in the stromal layer per section on 5.5 dpc was counted by ImageJ software (*n* = 5 for WT and *n* = 7 for *Lpar3* KO uteri). (E) CD31(+) area in the stromal layer per section in (D) was calculated by ImageJ software (*n* = 5), showing reduced angiogenesis in *Lpar3* KO pregnant uteri.

G, H  The expression of *Hbegf*, *Ptgs2*, *Bmp2*, and *Wnt4* were significantly lower in *Lpar3* KO pregnant uteri. (G) Representative ISH images of *Hbegf* and *Ptgs2* in *Lpar3* KO pregnant uteri on 4.0 dpc. Arrowheads indicate the position of embryos. (H) qRT–PCR quantification of *Hbegf*, *Bmp2* and *Wnt4* mRNAs in *Lpar3* KO pregnant uteri on 4.0 dpc (*Hbegf*, *n* = 7 for WT and *n* = 12 for *Lpar3* KO uteri) and 5.5 dpc (*Bmp2* and *Wnt4*, *n* = 7 for WT and *n* = 9 for *Lpar3* KO uteri).

Data information: Scale bar = 200 μm (A and D), 50 μm (F) and 100 μm (G). Each image is a representative from at least three independent experiments. Data are means + SEM, *$P < 0.05$, **$P < 0.01$, by Student's *t*-test.

Source data are available online for this figure.

Our findings may lead to new treatments for infertility. In *in vitro* fertilization (IVF), endometrial thickness is highly correlated with the success rate of implantation and pregnancy (Noyes *et al*, 1995; Lim & Wang, 2010). Interestingly, patients with recurrent implantation failure in IVF treatment have reduced LPA₃ levels (Achache *et al*, 2010). Thus, treatment of human females

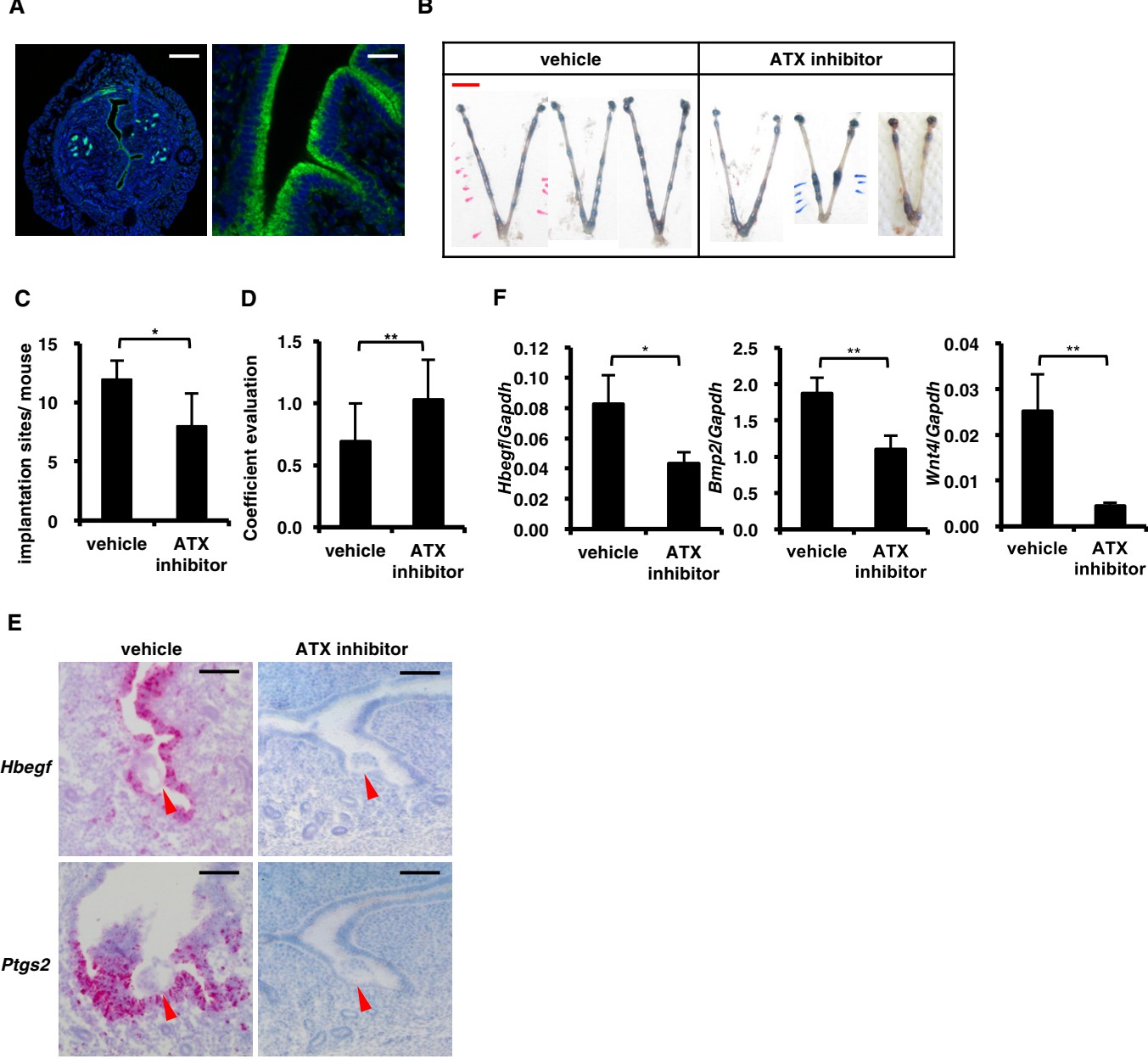

**Figure 7.  Autotaxin is responsible for LPA$_3$ in uteri during early pregnancy.**

A    Immunostaining of ATX in WT pregnant uteri at 3.5 dpc. The signal was highly detected in the luminal epithelium.

B–D    Embryo implantation was impaired by the inhibition of ATX like *Lpar3* KO uteri. (B) Representative photographs of pregnant uteri on 5.5 dpc from mice injected with ATX inhibitor at 3.5 dpc. (C and D) The number of implantation sites (C) and the degree of equidistance (D) in (B) (*n* = 7 for vehicle and *n* = 9 for ATX inhibitor).

E, F    The expression of *Hbegf*, *Ptgs2*, *Bmp2*, and *Wnt4* were significantly lower in pregnant uteri treated with the ATX inhibitor. (E) Representative ISH images of *Hbegf* and *Ptgs2* in pregnant uteri on 4.0 dpc injected with the ATX inhibitor at 3.5 dpc. Arrowheads indicate the position of embryos. (F) qRT–PCR quantification of *Hbegf*, *Bmp2*, and *Wnt4* mRNAs in pregnant uteri on 4.0 dpc (*Hbegf*, *n* = 8 for vehicle and *n* = 10 for ATX inhibitor) and 5.5 dpc (*Bmp2* and *Wnt4*, *n* = 15 for vehicle and *n* = 16 for ATX inhibitor) after treated with the ATX inhibitor at 3.5 dpc.

Data information: Each image in (A), (B), and (E) is a representative from at least three independent experiments. Scale bar: 200 μm (A, left), 20 μm (A, right), 1 cm (B) and 100 μm (E). Data are means + SEM, *$P < 0.05$, **$P < 0.01$, by Student's *t*-test.

with an LPA$_3$ agonist may increase the pregnancy rate by promoting the endometrial cell growth and increasing the endometrial thickness in IVF. This idea is supported by the facts that expression of *Lpar3* in mice is up- and down-regulated by P$_4$ and E$_2$,

respectively (Hama *et al*, 2006) and that the balance of female sex hormones is also critical for establishment of pregnancy in both species (Lim & Wang, 2010; Ramathal *et al*, 2010; Cha *et al*, 2012).

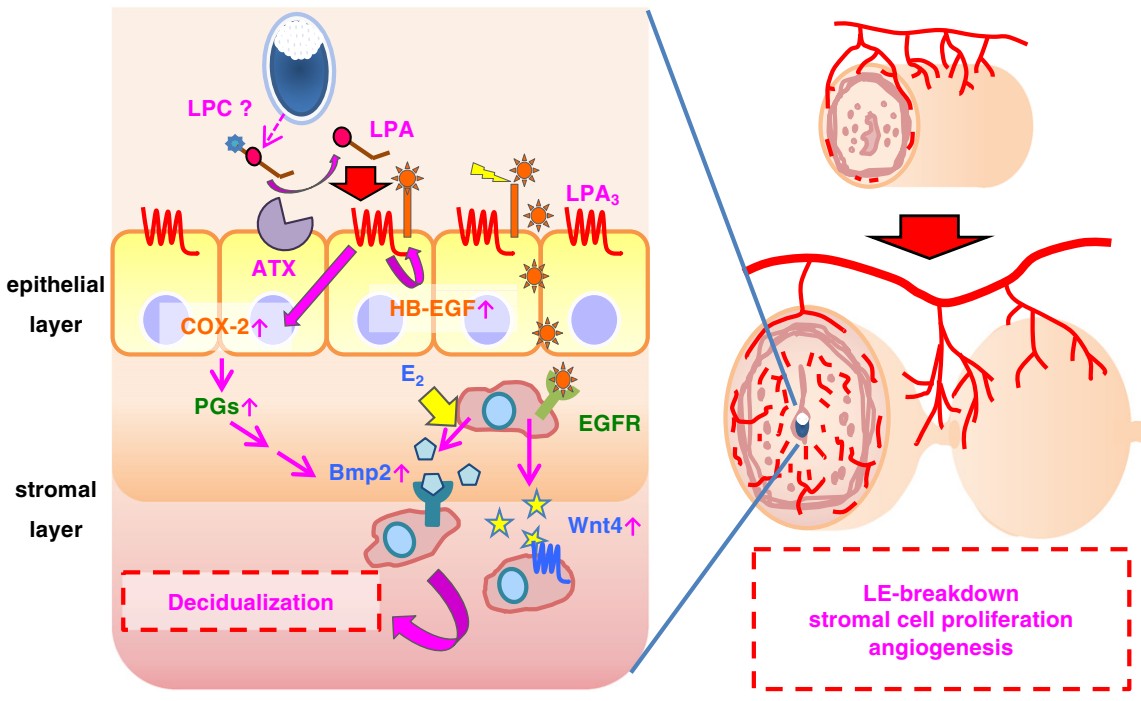

**Figure 8.  Proposed model for LPA-induced decidualization through ATX–LPA$_3$ axis.**

LPA is produced at the embryo-epithelial boundary in ATX-dependent manner and then activates epithelial LPA$_3$. Embryos are the possible source of LPC. LPA$_3$ activation induces heparin-binding EGF (HB-EGF) and cyclooxygenase-2 (COX-2) in the epithelial layer, which up-regulates Bmp2 and Wnt4 cooperatively with estrogen (E$_2$) signaling. Totally, LPA$_3$ signaling contributes LE-breakdown, angiogenesis, and cell proliferation in the stromal layer.

HB-EGF, COX-2, and Wnt4 have been implicated in the progression of endometriosis, the abnormal growth of endometrial tissues outside the uterus (Ota *et al*, 2001; Uno *et al*, 2010; Arosh *et al*, 2015; Miller *et al*, 2015). Interestingly, in our mouse endometriosis model, endometrial tissues from *Lpar3* KO mice were significantly less developed (Appendix Fig S6). Thus, LPA–LPA$_3$ signaling is critical in the development of endometriosis. Furthermore, HB-EGF, COX-2, and Wnt4 as well as Bmp2 have been shown to be risk factors for sex hormone-dependent diseases, such as prostatic hyperplasia, breast cancer, and ovarian cancer (Bentley *et al*, 1992; Ferrandina *et al*, 2002; Levin, 2003; Zong *et al*, 2012). Since LPA$_3$ is highly expressed in the prostate, mammary gland, and ovary (Bandoh *et al*, 1999), LPA$_3$ signaling might contribute to the progression of such diseases.

In summary, we propose a novel mechanism for LPA$_3$-mediated decidualization during the peri-implantation period as illustrated in Fig 8. (i) LPA is produced in the vicinity of the embryo in an ATX-dependent manner; (ii) LPA thus generated activates LPA$_3$ in the epithelial layer; (iii) activation of LPA$_3$ up-regulates HB-EGF and COX-2 in the epithelial layer; (iv) Both Bmp2 and Wnt4 are induced in the stromal layer via activation of EGFR, COX-2, and ERα, leading to the decidual reactions including LE-breakdown, stromal proliferation, and angiogenesis. Because expression of LPA$_3$ and LPA$_3$ signaling is highly dependent on female sex hormones (P$_4$ and E$_2$, respectively; Hama *et al*, 2006; Diao *et al*, 2015), this mechanism may operate not only in female reproduction but also in pathological diseases such as endometriosis and female reproductive cancers.

# Materials and Methods

### Reagents

LPA (1-oleoyl (18:1)) was purchased from Avanti Polar Lipids. A potent LPA$_3$ agonist T13 was synthesized as described previously (Tamaruya *et al*, 2004; Kano *et al*, 2008; Hama & Aoki, 2010). LPA and T13 were dried under nitrogen gas and dissolved in 0.01% fatty acid-free bovine serum albumin (Sigma-Aldrich)-PBS using water bath sonication and stocked in −20°C. COX-2 selective inhibitor (Celecoxib) was from TCI. EGFR inhibitor (AST1306), BMPR inhibitor (LDN193189), and Tankyrase inhibitor (XAV939) were obtained from Adooq Bioscience. Estrogen receptor (ERα) antagonist (fulvestrant) was obtained from Sigma-Aldrich. All inhibitors and the ERα antagonist were dissolved in DMSO and stored at −20°C. ATX inhibitor (S15-00826) was kindly donated by Shionogi pharmaceutical company. The selectivity toward ENPP family members and pharmacokinetics of the ATX inhibitor was shown in Fig EV7. Other chemicals were purchased from Wako Pure Chemical Industries unless otherwise indicated.

### Antibodies

Anti-autotaxin (ATX) polyclonal antibody (pAb) made in guinea pig was a kind gift from Drs Masahiko Watanabe and Masanori Tachikawa (Hokkaido University, Japan). The following antibodies were purchased from distributors: anti-mouse CD31 (PECAM)

monoclonal antibody (mAb) made in rat (BD Pharmingen); anti-BrdU mAb conjugated with fluorescein made in mouse (Roche); anti-mouse E-cadherin mAb made in Rabbit (CST); anti-guinea pig, anti-rat, anti-fluorescein secondary antibodies (biotinylated; Vector); anti-rabbit secondary antibody-Alexa488 (Invitrogen). ATX, CD31, and BrdU were detected using biotinylated secondary antibody followed by TSA Alexa Fluor 488 kit (Invitrogen).

## Mice

ICR mice were purchased from CLEA Japan. *Lpar3*$^{-/-}$ mice (ICR background) were generated by crossing C57BL/6J *Lpar3*$^{+/-}$ males with ICR wild-type females over eight times.

## TGF$\alpha$ shedding assay

TGF$\alpha$ shedding assay was performed as described previously (Inoue *et al*, 2012). Briefly, HEK293 cells were seeded in 12-well dishes at a density of $2 \times 10^5$ cells/dish and cultured at 37°C for 24 h. Then, the cells were transfected with cDNAs encoding alkaline phosphatase (AP)-tagged TGF$\alpha$ (AP-TGF$\alpha$, 0.25 μg), LPA receptors (each 0.1 μg) using Lipofectamine 2000 as transfection reagent. The cells were seeded in 96-well plate at a density of $4 \times 10^5$ cells/ml (90 μl) in HBSS containing 5 mM HEPES (pH 7.4). After incubation at 37°C for 30 min, the cells were stimulated with LPA or T13 and incubated at 37°C for 1 h. After 1 h, the cells centrifuged at 190 g for 3 min and supernatant (80 μl) was moved into a new 96-well plate. An amount of 80 μl of 10 mM *p*-nitrophenyl phosphate (*p*NPP) in 2× *p*NPP buffer [40 mM Tris–HCl (pH 9.5)], 40 mM NaCl and 10 mM MgCl2)/well was added to the supernatant and the cells, and measured at optical density at 405 nm (OD$_{405}$). After incubation at 37°C for 1 h, both the supernatant and the cells were measured at OD$_{405}$. TGF$\alpha$ shedding activity was calculated by OD$_{405}$ of 0 and 1 h.

## Intrauterine injection of T13 and inhibitors

Adult females (8–12 weeks) were mated with vasectomized males. The day a plug was found after mating was designed as 0.5 dpc of pseudopregnancy. At 3.5 dpc, females were anesthetized with three types of reagent cocktail (M/M/B; 0.3 mg/kg of medetomidine, 4.0 mg/kg of midazolam, and 5.0 mg/kg of butorphanol) *i.p.* and then subjected intrauterine injection, as previously described (Wu & Gu, 1981). Briefly, 5 μl/horn of T13 solution (1 mM) or 0.01% BSA-PBS was injected into the uterine cavity adjacent to the ovary. Inhibitors for EGFR, BMPR, or Tankyrase (100 μM of final concentration) were mixed with T13 solution before injection, respectively. ER$\alpha$ antagonist (100 μg/mouse in 100 μl DMSO) or COX-2 inhibitor (10 mg/mouse in 100 μl DMSO) was subcutaneously injected into mice 3 h before or just before T13 injection, respectively. The mice were intravenously injected with 200 μl of 1% Evans Blue solution to visualize decidualized area at 5.5 dpc.

## Oil infusion

Corn oil was infused intraluminally into the uterine horns as described previously (Chen *et al*, 2011) with some modifications. Briefly, 50 μl corn oil was infused at 3.5 dpc of pseudopregnancy. On 5.5 dpc, the mice were intravenously injected with 200 μl of 1%

Evans Blue solution to detect deciduoma, and uterine weights were recorded to assess the extent of decidualization.

## Evaluation of implantation in mice treated with autotaxin inhibitor

Adult female mice were mated with fertile WT males. At 3.5 dpc, females were injected with 5 μl/horn of autotaxin inhibitor (1 mM) or vehicle (3.3% DMSO in PBS) in a similar manner as T13 injection. On 5.5 dpc, mice were injected with 1% Evans Blue solution to visualize embryo implantation sites. The distance between each blastocyst was calculated by National Institutes of Health Image software. The coefficient of evaluation (= the degree of equidistance) was calculated by dividing the standard deviation by the mean distance between blastocysts in a horn.

## Immunohistochemistry

Uteri were fixed in 4% formaldehyde in phosphate-buffered saline (PBS) at 4°C overnight and embedded in O.C.T. compound (Sakura Finetek). Tissues were cryosectioned at 8 μm. The cryosections were plated on MAS-coated glass slides (Matsunami Glass), blocked in IgG from the same animal species of the secondary antibody for 30 min at room temperature, incubated at 4°C overnight with primary antibodies, incubated with biotinylated secondary antibody and TSA Alexa Fluor 488 kit (Invitrogen) for 5 min, counterstained with 4′,6-diamidino-2-phenylindole (DAPI, Sigma-Aldrich) to reveal the nuclei and photographed with a Zeiss LSM700 confocal fluorescence microscope. Number of nuclei and CD31-positive area were calculated by ImageJ (National Institutes of Health). For Figs 2A–C (upper panel), 5B, 6A and D, the brightness and contrast was changed using ImageJ (National Institutes of Health).

## Bromo-deoxyuridine (BrdU) *in vivo* incorporation assay

Females were injected with BrdU (100 mg per kg body weight) on the morning of 3.5 dpc (10:00). The mice were killed 2 h after injection, and the uteri were freshly embedded in O.C.T. compound (Sakura Finetek). The embedded uteri were cryosectioned, and the sections were fixed in methanol for 10 min at room temperature and immersed in 2N HCl at 37°C to denature DNA for immunohistochemical detection.

## Quantitative RT–PCR analysis

Total RNA from whole uterine tissues was isolated with a GenElute Mammalian Total RNA Miniprep kit (Sigma-Aldrich) and then reverse-transcribed with a High-Capacity cDNA RT Kits (Applied Biosystems) according to the manufacturer's instructions. PCRs were performed with SYBR Premix Ex Taq II (Takara Bio) and were monitored by ABI Prism 7300 (Applied Biosystems). Standard plasmids ranging from $10^3$ to $10^8$ copies per well were used to quantify the absolute number of transcripts of cDNA samples. The numbers of transcripts were normalized to the number of a house-keeping gene, *Gapdh*, in the same sample. PCR was performed using the following primers: 5′-TGACCCCAGCTCAGGGAAAG-3′ and 5′-GGGCTTAATCACCTGTTCAACTCTG-3′ for *Areg*; 5′-GATCTGTACCGCAGGCACTC-3′ and 5′-CCACGGCTTCTTCGTGATGG-3′ for *Bmp2*;

5′-ACTACAGGACTCGGAAGCAG-3′ and 5′-AAGGTTGGGGACAAGA AGCC-3′ for *Egf*; 5′-GACGCTGCTTTGTCTAGGTTC-3′ and 5′-CACACGGGGATCGTCTTCC-3′ for *Ereg*; 5′-AGGAGCGAGACCCCAC TAAC-3′ and 5′-CGGAGATGATGACCCTTTTG-3′ for *Gapdh*; 5′-TGCTGCCGTCGGTGATG-3′ and 5′-CAGACTCTCACCGGTCACCAA-3′ for *Hbegf*; 5′-AAGCGAGGACCTGGGTTCA-3′ and 5′-AAGGCGCA GTTTATGTTGTCTGT-3′ for *Ptgs2*; 5′-GCGAGCAATTGGCTGTAC-3′ and 5′-TTGTGTCACCACCTTCCCA-3′ for *Wnt4*.

### RNA *in situ* hybridization

Uteri were fixed overnight in 4% (w/v) paraformaldehyde at 4°C and then embedded in O.C.T. compound (Sakura Finetek). The tissues were cryosectioned at 8 μm and plated on MAS-coated glass slides (Matsunami Glass) and processed for RNA *in situ* detection using the RNAscope 2.0 High Definition RED kit according to manufacturer's instructions (ACDBio).

### LC-MS/MS analysis of prostaglandins and related lipids

T13-injected uteri were collected 9 h after the injection and processed for LC-MS-/MS-based lipidomics analyses as previously described (Arita, 2012). MS/MS analyses were performed in negative ion mode. Arachidonic acid-derived metabolites were identified and quantified by multiple reaction monitoring.

### LC-MS/MS analysis of LPA

To obtain a plasma sample, a blood sample was collected 1 h after the administration of ATX inhibitor (20 mg/kg, *i.p.*) and then centrifuged at 1,500 *g* for 5 min. Quantification of LPA was performed as previously described (Kato *et al*, 2016) using the highly sensitive LC-MS/MS system.

### Microarray analysis

The quantity and quality of total RNA were determined with an Agilent 2100 Bioanalyzer (RNA Nano 6000) and Nanodrop, respectively. For each of six time points (0.5, 1, 2, 24, 30, and 36 h after the T13 injection) and two conditions (vehicle control and T13-injected), total RNA was prepared from three individuals and pooled. DNA microarray analyses were carried out using Whole mouse genome oligo DNA microarray kit ver2.0 (4×44 K; Agilent Technology, for time points 0.5, 1, and 2 h) and GeneChip Mouse Gene 2.0 ST Array (Affymetrix, for time points 24, 30, and 36 h) according to the manufacturer's protocol. Up-regulated and down-regulated genes were defined as those whose expression levels were increased and decreased, respectively, by a factor of at least 2. Biological networks and pathways were analyzed with Ingenuity Pathway Analysis (IPA) software (Qiagen, CA, USA). The data have been deposited to the NCBI GEO database, with accession no. GSE87116 (0.5–2 h after the T13 injection) and GSE87161 (24–36 h after the T13 injection). The raw data of uterine-specific *Egfr* or *Wnt4* KO were kindly disclosed by the corresponding author of the previous article (Large *et al*, 2014). The data of *Bmp2* (Lee *et al*, 2007; accession no. GSE10193), *Egfr*, or *Wnt4* KO were analyzed using R (https://www.R-project.org/). Venn diagrams were created by Venny 2.1.0 (http://bioinfogp.cnb.csic.es/tools/venny/index.html).

### Pharmacokinetics of T13 and ATX inhibitor

T13 (10 nmol) was injected into uteri, and then, uteri were collected 0, 3, and 6 h after the injection. 20–30 mg of uterine tissues was homogenized in methanol. For the analysis of the ATX inhibitor, the inhibitor was injected to mice *p.o.* (10 or 30 mg/kg), and then, bloods were collected. Collected samples were processed for LC-MS/MS as previously performed (Okudaira *et al*, 2014).

### Phosphodiesterase activity assay

Phosphodiesterase activity was measured colorimetrically as previously described (Sakagami *et al*, 2005) with some modifications. Each mouse ENPP protein was transiently expressed in HEK293A cells using Lipofectamine 2000 as transfection reagent. For the analysis of the specificity of ATX inhibitor to ENPP proteins, culture supernatants were incubated with substrates, *p*-nitrophenyl thymidine monophosphate (4 mM, *p*NP-TMP, for ENPP1-5) or *p*-nitrophenylphosphorylcholine (4 mM, *p*NP-PC, for ENPP6, 7), in the presence or absence of the ATX inhibitor (1 μM) in a 96-well microplate at 37°C for 30 min. The amount of *p*-nitrophenolate (*p*-NP) was determined by the absorbance at 405 nm with a SpectraMax190 microplate reader (Molecular Devices). The phosphodiesterase activity was calculated assuming the inhibitor-free absorbance as 100% activity. For the determination of IC$_{50}$ value, ENPP2 protein was incubated with 1 mM *p*NP-TMP in the presence of ATX inhibitor for 40 min. A four-parameter sigmoid curve for IC$_{50}$ value was fitted to concentration−response plots using GraphPad Prism 6 (GraphPad, USA).

### Statistical analysis

The significance of differences between groups was determined by Student's *t*-test or a multiway analysis of variance (ANOVA) followed by a Bonferroni *post hoc* analysis using GraphPad Prism 6 (GraphPad).

### Study approval

Mice were maintained according to the Guidelines for Animal Experimentation of Tohoku University, and the protocol was approved by the Institutional Animal Care and Use Committee at Tohoku University (Approval number: 2014PhLMO-018).

Expanded View for this article is available online.

### Acknowledgements

We thank Shionogi pharmaceutical Co., Ltd for the gift of the ATX inhibitor. We also thank Dr Yasumasa Nishito (Core Technology and Research Center, Tokyo Metropolitan Institute of Medical Science) for DNA microarray analysis. The present work was supported partly by AMED-CREST (Japan Agency for Medical Research and Development, Core Research for Evolutional Science and Technology) for J.A. and M.M., PRESTO (Japan Science and Technology Agency, Precursory Research for Embryonic Science and Technology) for A.I., Ministry of Education, Culture, Sports, Science and Technology (MEXT) Grant-in-Aid for Scientific Research for J.A. and M.M.

## Author contributions

SA and KK designed the study and carried out most of the experiments. SA wrote the draft of the manuscript. JW, KK, and DS performed the LC-MS/MS experiments. TN, TF, MK, HI, and YY designed the endometriosis experiment and gave many useful comments. YH designed the experiments on decidual reactions. ST, YT, YS, and MM designed and carried out the experiment on DNA microarray. MA designed and performed mediator lipidomics experiment. JC provided the LPA₃ KO mice and gave many useful comments. JA supervised all aspects of the study, including experimental design, discussion, data interpretation, and modified the manuscript.

## Conflict of interest

The authors declare that they have no conflict of interest.

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
