## [Review Process File · The EMBO Journal]

Manuscript EMBO-2016-96290

Autotaxin-lysophosphatidic acid-LPA₃ signaling at the embryo-epithelial boundary controls decidualization pathways

Shizu Aikawa, Kuniyuki Kano, Asuka Inoue, Jiao Wang, Daisuke Saigusa, Takeshi Nagamatsu, Yasushi Hirota, Tomoyuki Fujii, Soken Tsuchiya, Yoshitaka Taketomi, Yukihiro Sugimoto, Makoto Murakami, Makoto Arita, Makoto Kurano, Hitoshi Ikeda, Yutaka Yatomi, Jerold Chun and Junken Aoki

Corresponding author: Junken Aoki, Tohoku University

Review timeline:	Submission date:	10 December 2016
	Editorial Decision:	03 February 2017
	Revision received:	13 March 2017
	Editorial Decision:	05 April 2017
	Revision received:	12 April 2017
	Accepted:	02 May 2017

Editor: Ieva Gailite

Transaction Report:

1st Editorial Decision

03 February 2017

Thank you for submitting your manuscript for consideration by the EMBO Journal. First, I would like to apologise for the undue delay in getting back to you with a decision, caused by delayed responses from the referees that had agreed to review the study - as it can unfortunately happen over the holiday season, despite multiple reminders sent from our office. In the meantime, we have now finally received all three reports on your manuscript, which I am copying below for your information.

As you can see from the comments, all three referees express interest in the presented mechanism of LPA generation in the uterus. However, they also raise substantive concerns with the analysis that would need to be addressed in order to consider publication here. I would like to invite you to submit your revised manuscript while addressing the comments of all three referees, and focusing in particular on the following points:

- Validation of the LPA₃ agonist and Autotaxin antagonist specificity and pharmacokinetics, as suggested by Referees #1 and #3. We support the comments of Referee #1 on the need for further characterisation of these reagents, but from our point of view, T13 data should not be excluded from the manuscript.

- Perform mass-spectrometry analysis of LPA species (Referees #1 and #3), or otherwise show autotaxin-induced LPA formation to strengthen the support for the proposed mechanism of LPA generation.
- Incorporate microarray data in the main text of the manuscript and provide a comparison with the data from similar studies, as requested by Referee #2.

I should add that it is The EMBO Journal policy to allow only a single major round of revision and that it is therefore important to resolve the main concerns raised at this stage.

When preparing your letter of response to the referees' comments, please bear in mind that this will form part of the Review Process File, and will therefore be available online to the community. For more details on our Transparent Editorial Process, please visit our website: http://emboj.emboress.org/about#Transparent_Process

We generally allow three months as standard revision time. Please contact us in advance if you would need an additional extension. As a matter of policy, competing manuscripts published during this period will not negatively impact on our assessment of the conceptual advance presented by your study. However, we request that you contact the editor as soon as possible upon publication of any related work to discuss how to proceed.

Please feel free to contact me if have any further questions regarding the revision. Thank you for the opportunity to consider your work for publication. I look forward to your revision.

REFEREE REPORTS

Referee #1:

This is an interesting paper reporting on the role of autotaxin (ATX, an LPA-generating phospholipase) and the LPA3 receptor in inducing decidualization in mouse embryos, with a focus on LPA3-mediated induced upregulation of HB-EGF and COX-2. Using a potent (but poorly characterized) LPA3 agonist (termed T13), Lpar(-/-) mice, microarray analysis, and an undefined ATX inhibitor, the authors find that the ATX-LPA-LPA3 signaling axis contributes to decidualization through upregulation of HB-EGF and COX-2 during placenta formation and embryonic development. These findings (together with previous studies), although not very novel, could be relevant for human endometriosis and premature birth.

The present study builds on previous work by the same group(s), where the authors analyzed defects in embryo implantation in Lpa3 KO mice. Generally, the experiments have been carefully performed, and comprise a lot of work that culminates in a relatively simple message. That being said, the present paper comes across as a collection of rather isolated observations. For me, the paper is a difficult read, not very reviewer friendly, since it lacks a coherent line of reasoning. In other words, it lacks sufficient focus.

Major points of concern

Interpretation of the results relies heavily on the use of an ill-defined pharmacological agonist (T13). The structure of T13 is not shown, while there is just one reference to T13 (ref. 23). But in the latter reference, 'T13' is not even mentioned (I must guess that T13 is LPA analogue #13 in ref. 23...?). I could not find published data on T13 pharmacokinetics in vivo. Without proper pharmacokinetic characterization, it is premature to use T13 for functional studies in mice.

A similar (but less severe) concern holds for the ATX inhibitor used. It has no name, no structure is shown, just a single reference to a Japanese patent (ref. 44, which does not belong in the reference list, in my opinion). To the best of my knowledge, there are several well-characterized ATX inhibitors commercially available. Why not using one of those..?

Although it is an unbiased approach, microarray analysis is not really necessary here, as it was done to discover the obvious, namely involvement of the usual ('classical') suspects HB-EGF and COX-2, which is of little novelty. In any case, the data shown in Tables S2 and S3 are not informative and should be deleted.

I am not sure if the LPA species analysis (Fig. 6G) is meaningful. The data refer to total tissue LPC, not extracellular LPC, right? In any case, the title of the legend to Fig. 6 (ATX is responsible for LPA production) is too strong. The authors do not show LPA production (or LPC hydrolysis, for that matter).

The signaling scheme of Fig. 7 is not easy to understand. Abbreviations and players involved should be explained in the legend.

The last paragraph of the Introduction should be rewritten. It is a difficult read. A simple take-home message is essential here in my opinion

Minor points that need to be addressed

English language and syntax need correction in several places.

The list of references could be more balanced. The authors should add relevant references where appropriate.

Finally,

My suggestion to the authors is to reconstruct the paper, by not focusing on the effects of T13, but rather on the analysis of Lpa3 KO mice. The authors may even consider publishing the T3 results separately, for instance in a more pharmacologically oriented journal.

Referee #2:

The manuscript by Aikawa and coworker investigates the role of the Autotaxin-LPA2 axis in the regulation of decidualization in the mouse uterus. They demonstrate that T13 an agonist of LPA induces decidualization in the mouse pseudo pregnant mouse uterus. This induction is dependent upon the LPA3 receptor baby utilizing knockout mice. They demonstrate that proliferation and vascularization is induced as well as HB-EGF Cox2 Wnt 4 and Bmp2. This pathway is not critical for decidualization as the LPA3KO mouse will decidualize with oil but may be able to override these pathways. All in all the findings in this paper are important and worthwhile. The one weakness is that the focus of HB-EGF is weak as ablation of HB EGF mice can decidualize. The interesting information in this paper is the microarray analysis in response to T13, however this data is buried in the Supplemental Data. This data should be put in the body of the paper and mined against other microarray data from other groups to show which pathways are regulated by T13. This data would make the manuscript exciting and of high interest.

Referee #3:

This paper deals with the control of a key event in placental formation and fetal development, that of decidualization. It identifies autotaxin and signaling through LPA3 receptors as key events in this process. Knockdown of Lpar3 or inhibition of autotaxin downregulated HB-EGF and COX-2 near to the embryos and attenuated decidual reactions. Conversely, activation of LPA3 using a selective agonist increased HB-EGF and COX-2. The work on the whole is detailed and convincing and the paper is well written. The paper is a natural extension of work published by the Ye et al. paper 2005 (ref. 21), which shows that LPA3, COX-2 and prostaglandins are required for implantation and embryo spacing.

Specific points

1. Supplementary figure 1. The cell type or system used in this figure and how this experiment was performed is not clear from the legend.
2. Figures 1, 2A, 4B,. These results should be quantified in addition to the depiction of the uteri and the histochemistry.

3. Fig. S6. Could the authors comment on the some of results in this figure that are shown as non-significant? Is this simply a result of variability and the number of experiments performed?
4. Fig. 6. This reviewer cannot find sufficient details of the identity of the autotaxin inhibitor nor the concentration used. It is essential that this information be provided before publication even if the patent number is provided to identify the compound.
5. Although strong evidence is provided to support the involvement of LPA3 receptors, perhaps the authors could discuss if LPA4 and LPA5 could be involved, especially since the T13 agonist shows activity against both of these other receptors? How can the T13 agonist be described as selective?
6. LC-MS/MS analysis of LPC is described but were LPA species measured, which are more important?
7. Decidualization in the mouse occurs "only in the vicinity of the embryos". However in the human there is a process called pre-decidualization that occurs in the absence of the embryo, i.e. the embryo (which in the mouse produces LPC) is not needed for pre-decidualization process, which is presumably in preparation should the human egg be fertilized and begin the process of implanting. The authors do not discuss how this pre-decidualization process differs from that of the mouse and how their findings in the mouse relate to this process and how this process of pre-decidualization might be "reinforced" by embryo implantation. This should be discussed.

1st Revision - authors' response

13 March 2017

Editor:

- *Validation of the LPA3 agonist and Autotaxin antagonist specificity and pharmacokinetics, as suggested by Referees #1 and #3. We support the comments of Referee #1 on the need for further characterisation of these reagents, but from our point of view, T13 data should not be excluded from the manuscript.*

Thank you for all the helpful comments. As requested, we determined the specificity of LPA₃ agonist (Fig EV1A) and ATX inhibitor (antagonist) (Fig EV1B). In addition, we performed pharmacokinetical analyses of both compounds (Fig EV1C, D).

- *Perform mass-spectrometry analysis of LPA species (Referees #1 and #3), or otherwise show autotaxin-induced LPA formation to strengthen the support for the proposed mechanism of LPA generation.*

Thank you for your comment. Detection of LPA is a very important issue of the present study. We tried to detect LPA both in blastocysts and uterine luminal fluids (isolated from uteri during the peri-implantation period) by our LC-MS/MS system specialized for lysophospholipids (Okudaira *et al*, 2014). While we could not detect LPA in the eggs, small amount of LPA (0.1-0.2 nM) was found in the uterine flushing fluids from the pregnant mice (Appendix Fig S3 in the revised manuscript). Interestingly, LPA with an unsaturated fatty acid (oleic or linoleic acid), a potent ligand for LPA₃ (Bandoh *et al*, 2000), was detected when the uteri were flushed with the saline containing albumin which is capable of extracting lysophospholipids from outer leaflet of the cells (Okudaira *et al*, 2014) (Appendix Fig S3 in the revised manuscript). LPA was hardly recovered in the albumin-free flushing fluids (Appendix Fig S3 in the revised manuscript), indicating clearly that LPA is present in the extracellular milieu. The concentration of LPA detected in the flushing fluids was too low to activate LPA₃ (LPA₃ can be activated by >100 nM of LPA: see also Fig EV1B in the revised manuscript). However, the estimated egg volume is $\sim 6 \times 10^{-14} \text{ m}^3$ (provided that the diameter of the egg is 50 μm), while the volume of uterine cavity is $\sim 5 \times 10^{-9} \text{ m}^3$: *i.e.* the approximate ratio of them = 1:10⁵. Assuming that LPA is produced only in the embryo-epithelial boundary, we can estimate that a high concentration of LPA enough to activate LPA₃ (normally μM order) is present there.

The present study also clearly showed that LPA₃ is specifically activated in the vicinity of the embryo, suggesting that LPA is present there. However, we could not show the local distribution of

LPA in the vicinity of the embryo. Thus, we changed the title of the Fig 7 “Autotaxin is responsible for LPA₃ activation in uteri during early pregnancy”.

We also added the discussion about LPA production in “Discussion” (Page 10, Line2-26 in the revised manuscript).

- Incorporate microarray data in the main text of the manuscript and provide a comparison with the data from similar studies, as requested by Referee #2.

Thank you for the helpful comment. According to the suggestion, we incorporated the microarray data in the main text (Table1-4 in the revised manuscript) and compared the data with the data from previous related studies (Large *et al*, 2014) (Fig 4A in the revised manuscript). We also described the detail results in Page7, Line14-25 in the revised manuscript.

Referee #1:

Major points of concern

Interpretation of the results relies heavily on the use of an ill-defined pharmacological agonist (T13). The structure of T13 is not shown, while there is just one reference to T13 (ref. 23). But in the latter reference, 'T13' is not even mentioned (I must guess that T13 is LPA analogue #13 in ref. 23...?). I could not find published data on T15 pharmacokinetics in vivo. Without proper pharmacokinetic characterization, it is premature to use T13 for functional studies in mice.

Thank you for the comments. First, we showed the structure of T13 in Fig EV1A in the revised manuscript. We also added two references describing T13 (Hama and Aoki, 2010; Kano *et al*, 2008). We performed pharmacokinetic analysis of T13 in uteri (Fig EV1B in the revised manuscript), showing that T13 was present in the uterine cavity at least for 3-6 hours after the intrauterine injection.

A similar (but less severe) concern holds for the ATX inhibitor used. It has no name, no structure is shown, just a single reference to a Japanese patent (ref. 44, which does not belong in the reference list, in my opinion). To the best of my knowledge, there are several well-characterized ATX inhibitors commercially available. Why not using one of those..?

Thank you for the critical comments. In collaboration with a pharmaceutical company, we recently developed a series of ATX inhibitors (S series) which showed more potent inhibitory activities than previously well-characterized ATX inhibitor, *e.g.* HA130. In addition, for *in vivo* use, much amount of compounds are needed. Thus, in this study, we utilized one of the potent S series compounds named S15-00826. As the editor requested, we determined the specificity of S15-00826 using several recombinant enzymes belonging to the ENPP family (ENPP1-7, ATX is known as ENPP2) (Fig EV7A in the revised manuscript). We also performed the pharmacokinetics of the compound (Fig EV7B in the revised manuscript). Due to a possible patent problem, we couldn't disclose the structure of S15-00826.

Although it is an unbiased approach, microarray analysis is not really necessary here, as it was done to discover the obvious, namely involvement of the usual ('classical') suspects HB-EGF and COX-2, which is of little novelty. In any case, the data shown in Tables S2 and S3 are not informative and should be deleted.

Thank you for the helpful comments. Other reviewers and the editor were positive for the microarray analysis. Thus, we decided not to delete the data from the manuscript.

I am not sure if the LPA species analysis (Fig. 6G) is meaningful. The data refer to total tissue LPC, not extracellular LPC, right? In any case, the title of the legend to Fig. 6 (ATX is responsible for LPA production) is too strong. The authors do not show LPA production (or LPC hydrolysis, for that matter).

Thank you for the helpful comments. Yes, we measured the total LPC in the embryo tissues. Thus, the LPC detected is not necessarily present extracellularly. LPC is always detected in cells, where LPC is distributed both intra- and extracellularly. Thus, from Fig 6G (Appendix Fig S4 in the

revised manuscript), we can speculate that embryo tissue is a possible source of LPC. Accordingly, as suggested, we changed the title "Autotaxin is responsible for LPA₃ activation in uteri during early pregnancy". We also added the discussion about LPA production in "Discussion" (Page 10, Line 2-26 in the revised manuscript).

The signaling scheme of Fig. 7 is not easy to understand. Abbreviations and players involved should be explained in the legend.

Thank you for the suggestion. We changed the legend of Fig 7 (Fig 8 in the revised manuscript).

The last paragraph of the Introduction should be rewritten. It is a difficult read. A simple take-home message is essential here in my opinion

Thank you for the suggestion. We rewrote this part and made it easier to read as much as possible (Page 5, Line 17-26 in the revised manuscript).

Minor points that need to be addressed

English language and syntax need correction in several places.

Thank you for pointing out. As suggested, our revised manuscript was checked by a native English speaker.

The list of references could be more balanced. The authors should add relevant references where appropriate.

As suggested, we balanced the references as much as possible. Newly added references are highlighted in yellow.

Finally,

My suggestion to the authors is to reconstruct the paper, by not focusing on the effects of T13, but rather on the analysis of Lpa3 KO mice. The authors may even consider publishing the T13 results separately, for instance in a more pharmacologically oriented journal.

Thank you for the helpful comments. We understand the point. However, the editor suggested not to exclude the T13 data from the manuscript, we follow the instruction of the editor.

Referee #2:

The interesting information in this paper is the microarray analysis in response to T13, however this data is buried in the Supplemental Data. This data should be put in the body of the paper and mined against other microarray data from other groups to show which pathways are regulated by T13. This data would make the manuscript exciting and of high interest.

Thank you for the helpful comments. As suggested, we moved the supplemental tables about microarray analysis to the main Tables (Table 1-4 in the revised manuscript).

We also compared our data with the microarray data of uteri null for either *Egfr*, *Bmp2* and *Wnt4* under the decidual stimuli (Large *et al*, 2014). We found a negative correlation in the expression pattern between T13-injected uteri and these KO uteri (the venn diagram is put on Fig 4A in the revised manuscript), suggesting that LPA₃ induces activation of EGFR and Bmp2/Wnt4 signaling. This result strengthened our conclusion which LPA₃ activation induces HB-EGF and Bmp2/Wnt4, contributing to decidualization. We also described the detail results in Page 7, Line 14-25 in the revised manuscript.

Referee #3:

1. Supplementary figure 1. The cell type or system used in this figure and how this experiment was performed is not clear from the legend.

Thank you for the comment. As suggested, we described the method in detail in the legend (Fig EV1 in the revised manuscript).

2. *Figures 1, 2A, 4B. These results should be quantified in addition to the depiction of the uteri and the histochemistry.*

Thank you for the comment. We quantified the images and the quantified data are added to the Figures (Fig 1B, Fig 2A and Fig 5B in the revised manuscript).

3. *Fig. S6. Could the authors comment on the some of results in this figure that are shown as non-significant? Is this simply a result of variability and the number of experiments performed?*

Thank you for the suggestion. In this figure (Appendix FigS2 in the revised manuscript), we just want to show that the inhibition of EGFR, COX-2 or ERA didn't decrease the expressions of *Hbegf* and *Ptgs2*. As referee #2 pointed out, it is not important to show the statistical significance here. Accordingly, we deleted the SD from the graph.

4. *Fig. 6. This reviewer cannot find sufficient details of the identity of the autotaxin inhibitor nor the concentration used. It is essential that this information be provided before publication even if the patent number is provided to identify the compound.*

Thank you for the critical comments. In collaboration with a pharmaceutical company, we recently developed a series of ATX inhibitors (S series) which showed more potent inhibitory activities than previously well-characterized ATX inhibitor, *e.g.* HA130. In addition, for *in vivo* use, much amount of compounds are needed. Thus, in this study, we utilized one of the potent S series compounds named S15-00826. As the editor requested, we determined the specificity of S15-00826 using several recombinant enzymes belonging to the ENPP family (ENPP1-7, ATX is known as ENPP2) (Fig EV7A in the revised manuscript). We also performed the pharmacokinetics of the compound (Fig EV7B in the revised manuscript). Due to a possible patent problem, we couldn't disclose the structure of S15-00826. We also added the concentration of S15-00826 for the intrauterine injection in "Materials and Methods" (Page15, Line7-14 in the revised manuscript).

5. *Although strong evidence is provided to support the involvement of LPA3 receptors, perhaps the authors could discuss if LPA4 and LPA5 could be involved, especially since the T13 agonist shows activity against both of these other receptors? How can the T13 agonist be described as selective?*

Thank you for the helpful comments. As Referee#2 said, T13 has high activities against not only LPA3 but also LPA4 and LPA5. However, these two receptors, LPA4 and LPA5, are almost absent in uterus during peri-implantation period (Ye *et al*, 2011). In addition, knockout of them in female mice didn't show any problems in early pregnancy events (Sumida *et al*, 2010; Lin *et al*, 2012). These facts strongly support that T13 selectively activates LPA3 in uterus, inducing decidual events.

6. *LC-MS/MS analysis of LPC is described but were LPA species measured, which are more important?*

Thank you for your comment. Detection of LPA is a very important issue of the present study. We tried to detect LPA both in blastocysts and uterine luminal fluids (isolated from uteri during the peri-implantation period) by our LC-MS/MS system specialized for lysophospholipids (Okudaira *et al*, 2014). While we could not detect LPA in the eggs, small amount of LPA (0.1-0.2 nM) was found in the uterine flushing fluids from the pregnant mice (Appendix Fig S3 in the revised manuscript). Interestingly, LPA with an unsaturated fatty acid (oleic or linoleic acid), a potent ligand for LPA₃ (Bandoh *et al*, 2000), was detected when the uteri were flushed with the saline containing albumin which is capable of extracting lysophospholipids from outer leaflet of the cells (Okudaira *et al*, 2014) (Appendix Fig S3 in the revised manuscript). LPA was hardly recovered in the albumin-free flushing fluids (Appendix Fig S3 in the revised manuscript), indicating clearly that LPA is present in the extracellular milieu. The concentration of LPA detected in the flushing fluids was too low to activate LPA₃ (LPA₃ can be activated by >100 nM of LPA: see also Fig EV1B in the revised manuscript). However, the estimated egg volume is $\sim 6 \times 10^{-14} \text{ m}^3$ (provided that the diameter of the egg is 50 μm), while the volume of uterine cavity is $\sim 5 \times 10^{-9} \text{ m}^3$: *i.e.* the approximate ratio of them

= 1:10⁵. Assuming that LPA is produced only in the embryo-epithelial boundary, we can estimate that a high concentration of LPA enough to activate LPA₃ (normally μM order) is present there.

The present study also clearly showed that LPA₃ is specifically activated in the vicinity of the embryo, suggesting that LPA is present there. However, we could not show the local distribution of LPA in the vicinity of the embryo. Thus, we changed the title of the Fig 7 “Autotaxin is responsible for LPA₃ activation in uteri during early pregnancy”.

We also added the discussion about LPA production in “Discussion” (Page 10, Line2-26 in the revised manuscript).

7. Decidualization in the mouse occurs "only in the vicinity of the embryos". However in the human there is a process called pre-decidualization that occurs in the absence of the embryo, i.e. the embryo (which in the mouse produces LPC) is not needed for pre-decidualization process, which is presumably in preparation should the human egg be fertilized and begin the process of implanting. The authors do not discuss how this pre-decidualization process differs from that of the mouse and how their findings in the mouse relate to this process and how this process of pre-decidualization might be "reinforced" by embryo implantation. This should be discussed.

Thank you for the helpful comment. As pointed out, there are some species differences in the decidual processes. However, the expression of LPA₃ in female reproductive tissues is conserved in mammals. Indeed, in mouse, sheep, cow and human, LPA₃ was expressed in the uterine epithelial layer in a female sex hormone-dependent manner (Guo *et al*, 2013; Hama *et al*, 2006; Kamińska *et al*, 2008; Liszewska *et al*, 2012). In addition, ATX and LPA were detected in the reproductive biological fluids such as follicular fluids and uterine luminal fluids including human samples (Liszewska *et al*, 2009; Seo *et al*, 2012; Yamamoto *et al*, 2016). Thus, LPA₃ appears to regulate the female reproductive systems in wide range of mammalian species including human, although there are some slight differences in the process of decidualization between species as Referee #3 pointed out. We added such discussion in “Discussion” (Page11, Line8-16 in the revised manuscript).

References

- Bandoh K, Aoki J, Taira A, Tsujimoto M, Arai H, Inoue K (2000) Lysophosphatidic acid (LPA) receptors of the EDG family are differentially activated by LPA species. Structure-activity relationship of cloned LPA receptors. *FEBS Lett* 478: 159-65
- Hama K, Aoki J, Bandoh K, Inoue A, Endo T, Amano T, Suzuki H, Arai H (2006) Lysophosphatidic receptor, LPA3, is positively and negatively regulated by progesterone and estrogen in the mouse uterus. *Life Sci* 79: 1736-40
- Hama K, Aoki J (2010) LPA(3), a unique G protein-coupled receptor for lysophosphatidic acid. *Prog Lipid Res* 49: 335-42
- Kano K, Arima N, Ohgami M, Aoki J (2008) LPA and its analogs-attractive tools for elucidation of LPA biology and drug development. *Curr Med Chem* 15: 2122-31
- Large MJ, Wetendorf M, Lanz RB, Hartig SM, Creighton CJ, Mancini MA, Kovanci E, Lee KF, Threadgill DW, Lydon JP, Jeong JW, DeMayo FJ (2014) The epidermal growth factor receptor critically regulates endometrial function during early pregnancy. *PLoS Genet* 10: e1004451
- Okudaira M, Inoue A, Shuto A, Nakanaga K, Kano K, Makide K, Saigusa D, Tomioka Y, Aoki J (2014) Separation and quantification of 2-acyl-1-lysophospholipids and 1-acyl-2-lysophospholipids in biological samples by LC-MS/MS. *J Lipid Res* 55: 2178-92
- Lin ME, Rivera RR, Chun J (2012) Targeted deletion of LPA5 identifies novel roles for lysophosphatidic acid signaling in development of neuropathic pain. *J Biol Chem* 287: 17608-17
- Sumida H, Noguchi K, Kihara Y, Abe M, Yanagida K, Hamano F, Sato S, Tamaki K, Morishita Y, Kano MR, Iwata C, Miyazono K, Sakimura K, Shimizu T, Ishii S (2010) LPA4 regulates blood and lymphatic vessel formation during mouse embryogenesis. *Blood* 116: 5060-70
- Ye X, Herr DR, Diao H, Rivera R, Chun J (2011) Unique uterine localization and regulation may differentiate LPA3 from other lysophospholipid receptors for its role in embryo implantation. *Fertil Steril* 95: 2107-13, 2113.e1-4
- Guo H, Gong F, Luo KL, Lu GX (2013) Cyclic regulation of LPA3 in human endometrium. *Arch Gynecol Obstet* 287: 131-8
- Kamińska K, Wasielek M, Bogacka I, Blitek M, Bogacki M (2008) Quantitative expression of lysophosphatidic acid receptor 3 gene in porcine endometrium during the periimplantation period and estrous cycle. *Prostaglandins Other Lipid Mediat* 85: 26-32

- Liszewska E, Reinaud P, Billon-Denis E, Dubois O, Robin P, Charpigny G (2009) Lysophosphatidic acid signaling during embryo development in sheep: involvement in prostaglandin synthesis. *Endocrinology* 150: 422-34
- Liszewska E, Reinaud P, Dubois O, Charpigny G (2012) Lysophosphatidic acid receptors in ovine uterus during estrous cycle and early pregnancy and their regulation by progesterone. *Domest Anim Endocrinol* 42: 31-42
- Seo H, Choi Y, Shim J, Kim M, Ka H (2012) Analysis of the lysophosphatidic acid-generating enzyme ENPP2 in the uterus during pregnancy in pigs. *Biol Reprod* 87: 77
- Yamamoto J, Omura M, Tuchiya K, Hidaka M, Kuwahara A, Irahara M, Tanaka T, Tokumura A (2016) Preferable existence of polyunsaturated lysophosphatidic acids in human follicular fluid from patients programmed with in vitro fertilization. *Prostaglandins Other Lipid Mediat* 126: 16-23

2nd Editorial Decision

05 April 2017

Thank you for submitting a revised version of your manuscript. The manuscript has now been seen by two of the original referees. While referee #3 finds that their concerns have been sufficiently addressed, referee #1 points out that the requested characterisation of autotaxin inhibitor has not been provided. I agree with referee #1 that this information should be added to the manuscript before it can be accepted for publication here.

Therefore, I would like to invite you to submit a revised manuscript, addressing the following technical and editorial issues:

- 1) Please provide data on autotaxin inhibitor validation.
- 2) Please add "region of interest" boxes to mark magnified areas in the upper panels of Figure 2C and left panels of figure 3B.
- 3) The magnified panel in Hbegf vehicle condition in Figure 3B is not correctly rotated in comparison to the lower magnification panel, please adjust.
- 4) There is a reference to Figure 1C on the page 6, while the Figure 1C panel has been removed in the revised version. Please correct in the manuscript text.
- 5) Please include the exact number of replicates (instead of a range, e.g. n = 4-11) used per experiment in the figure legends.
- 6) Immunofluorescence signal in panels in Figures 2A, 2B, 2C (upper panel), 5B, 6A and 6D is unfortunately weak, and will not be visible in printed form. Please let me know if you have any suggestions how to address this. One possibility would be to increase the contrast and submit the unmodified images as source data.

When preparing your letter of response to the referees' comments, please bear in mind that this will form part of the Review Process File, and will therefore be available online to the community. For more details on our Transparent Editorial Process, please visit our website: http://emboj.embopress.org/about#Transparent_Process

Please feel free to contact me if have any further questions regarding the revision. Thank you again for giving us the chance to consider your manuscript for The EMBO Journal. I am looking forward to seeing the final revised version.

REFEREE REPORTS

Referee #1:

In the revised version of their manuscript, the authors have adequately addressed many of my points of concern, but not all. One remaining problem is the use of an unpublished and chemically undefined ATX inhibitor. A concern also raised by Referee #3. There is no structure shown because of patent issues, which I find hard to accept for a publication in EMBO J. Drug potency is not determined, while the compound even lacks a name (in the Results text and figure legends). The use of this compound is the more surprising since there are several well-defined and much better characterized ATX inhibitors available, as I mentioned in my previous report. These include PF-8380 and ONO-8430506, which the authors have used in a recent study (Aikawa et al., *BBRC* 2017). At the very least, the authors should determine the in vitro potency of their novel compound

(IC₅₀ curves), and how the compound affects circulating LPA levels. The reader should be able to judge how this new ATX inhibitor compares to the established ones.

Referee #3:

The answers to the original criticisms are satisfactory.

2nd Revision - authors' response

12 April 2017

Editor

1) Please provide data on autotaxin inhibitor validation.

Thank you for your comment. As suggested, we revised the figure EV7 and show the chemical structure (Figure EV7A), name and the IC₅₀ value (curve) (Figure EV7B) of the ATX inhibitor (S15-00826) used in this study in addition to the data on the specificity (Figure EV7C in the revised manuscript) and pharmacokinetics (Figure EV7D in the revised manuscript). We also show the effect of the inhibitor on circulating LPA in mice (Figure EV7E). Accordingly, we revised Materials and Methods (Page 18, Line 23-26 in the revised manuscript) and the legend of EV7 (Page 34, Line 17-27 in the revised manuscript). The IC₅₀ value for S15-00826 is ~38 nM in ATX assay using *p*-nitrophenyl TMP as a substrate. The IC₅₀ values for PF-8380 and ONO-8430506 were reported to be ~2.8 nM and ~10 nM, although these values were determined using different assay systems (different substrate and concentration).

2) Please add "region of interest" boxes to mark magnified areas in the upper panels of Figure 2C and left panels of figure 3B.

Thank you for your comment. We added "region of interest" boxes to each panel. Accordingly, we added the description about the boxes in each legend (yellow-highlighted in the revised manuscript).

3) The magnified panel in Hbegf vehicle condition in Figure 3B is not correctly rotated in comparison to the lower magnification panel, please adjust.

Thank you for your comment. We rotated the panel you pointed out.

4) There is a reference to Figure 1C on the page 6, while the Figure 1C panel has been removed in the revised version. Please correct in the manuscript text.

Thank you for your comment. We removed the reference to Fig 1C from the text.

5) Please include the exact number of replicates (instead of a range, e.g. $n = 4-11$) used per experiment in the figure legends.

Thank you for your comment. We added the exact number of replicates to each figure legend (Figure 1-6), which are highlighted in yellow color in the revised manuscript. We also apologized that we described $n = 4-11$ as the number of replicates in the legend of Fig 3A, but actually $n = 4-12$ is correct. We corrected the number in the revised manuscript.

6) Immunofluorescence signal in panels in Figures 2A, 2B, 2C (upper panel), 5B, 6A and 6D is unfortunately weak, and will not be visible in printed form. Please let me know if you have any suggestions how to address this. One possibility would be to increase the contrast and submit the unmodified images as source data.

Thank you for your comments. According to your suggestion, we changed the brightness and contrast of each pictures and submitted the source data. Accordingly, we added the method for changing the brightness and contrast into "Materials and Methods" in the revised manuscript (Page 15, Line 26-27).

Referee #1:

In the revised version of their manuscript, the authors have adequately addressed many of my points of concern, but not all. One remaining problem is the use of an unpublished and chemically undefined ATX inhibitor. A concern also raised by Referee #3. There is no structure shown because of patent issues, which I find hard to accept for a publication in EMBO J. Drug potency is not determined, while the compound even lacks a name (in the Results text and figure legends). The use of this compound is the more surprising since there are several well-defined and much better characterized ATX inhibitors available, as I mentioned in my previous report. These include PF-8380 and ONO-8430506, which the authors have used in a recent study (Aikawa et al., BBRC 2017). At the very least, the authors should determine the in vitro potency of their novel compound (IC₅₀ curves), and how the compound affects circulating LPA levels. The reader should be able to judge how this new ATX inhibitor compares to the established ones.

Thank you for your comment. As suggested, we revised the figure EV7 and show the chemical structure (Figure EV7A), name and the IC₅₀ value (curve) (Figure EV7B) of the ATX inhibitor (S15-00826) used in this study in addition to the data on the specificity (Figure EV7C in the revised manuscript) and pharmacokinetics (Figure EV7D in the revised manuscript). We also show the effect of the inhibitor on circulating LPA in mice (Figure EV7E). Accordingly, we revised Materials and Methods (Page 18, Line 23-26 in the revised manuscript) and the legend of EV7 (Page 34, Line 17-27 in the revised manuscript). The IC₅₀ value for S15-00826 is ~38 nM in ATX assay using *p*-nitrophenyl TMP as a substrate. The IC₅₀ values for PF-8380 and ONO-8430506 were reported to be ~2.8 nM and ~10 nM, although these values were determined using different assay systems (different substrate and concentration).

Referee #3:

The answers to the original criticisms are satisfactory.
Thank you for your review and evaluation for us.

3rd Editorial Decision

02 May 2017

Thank you for submitting a revised version of your manuscript. It has now been seen by one of the original referees, who finds that all criticisms have been sufficiently addressed and recommends the manuscript for publication. I am now pleased to inform you that your manuscript has been accepted for publication in the EMBO Journal.

Corresponding Author Name: Junken Aoki

Journal Submitted to: The EMBO Journal

Manuscript Number: EMBOJ-2016-96290